# Data fusion uncertainty-enabled methods to map street-scale hourly NO$_2$ in Barcelona city: a case study with CALIOPE-Urban v1.0

Alvaro Criado[1], Jan Mateu Armengol[1], Hervé Petetin[1], Daniel Rodriguez-Rey[1], Jaime Benavides[1,3], Marc Guevara[1], Carlos Pérez García-Pando[1,2], Albert Soret[1], and Oriol Jorba[1]

[1]Barcelona Supercomputing Center, Barcelona, Spain
[2]ICREA, Catalan Institution for Research and Advanced Studies, Barcelona, Spain
[3]Department of Environmental Health Sciences, Mailman School of Public Health, Columbia University, New York, NY 10032, USA

**Correspondence:** Alvaro Criado (alvaro.criado@bsc.es) & Jan Mateu Armengol (jan.mateu@bsc.es)

**Abstract.** Comprehensive monitoring of NO$_2$ exceedances is imperative for protecting human health, especially in trafficked urban areas. However, accurate spatial characterization of exceedances is challenging due to the typically low density of air quality monitoring stations and the inherent uncertainties of urban air quality models. We study how observational data from different sources and time scales can be combined with a dispersion air quality model to obtain bias-corrected NO$_2$ hourly maps at the street scale. We present a kriging-based data-fusion workflow that merges a dispersion model output with continuous hourly observations, and uses a machine-learning-based Land Use Regression (LUR) model constrained with past short intensive passive dosimeter campaigns observations. While the hourly observations allow to bias-adjust the temporal variability of the dispersion model, the microscale-LUR model adds information on the NO$_2$ spatial patterns. Our method includes uncertainty calculation based on the estimated error variance of the Universal Kriging technique, which is subsequently used to produce urban maps of probability of exceeding the 200 $\mu g/m^3$ hourly and the 40 $\mu g/m^3$ NO$_2$ annual average limits. We assess the statistical performance of this approach in the city of Barcelona for the year 2019. Our results show that simply merging the monitoring stations with the model output already significantly increases the correlation coefficient (*r*) by +29 % and decreases the Root Mean Square Error (*RMSE*) by -32 %. When adding the time-invariant LUR model in the data-fusion workflow, the improvement is even more remarkable: +46 % and -48 % for the *r* and *RMSE*, respectively. Our work highlights the usefulness of high-resolution spatial information in data-fusion methods to estimate exceedances at the street scale better.

## 1 Introduction

Air pollution is the leading environmental risk factor globally (WHO, 2021). Mortality, the decrease in life quality, and the detrimental economic effects associated with air pollution are pressing decision-makers to take action, especially in urban areas, where more than 50 % of the global population lives and air quality standards are frequently exceeded. In the city of Barcelona (Spain), the high vehicle density (about 5800 vehicles $km^{-2}$ (Rivas et al., 2014)) induces a chronic NO$_2$ problem, which makes Barcelona the sixth European city with the highest mortality associated with NO$_2$ exposure (ISGlobal, 2021; Khomenko et al.,

2021). In this context, obtaining information on high-resolution exposure to $NO_2$ is crucial for decision-making in urban air quality management.

During the last decades, several approaches have been developed to estimate $NO_2$ exposure at different spatio-temporal scales (Denby, 2011). A common one is the Land Use Regression (LUR) model, which relates explanatory variables of different nature (land use cover, population density, traffic, climate, and others) with air quality observations using regression models (Briggs et al., 1997; Hoek et al., 2008; Beelen et al., 2013). LUR models are generally skillful, relatively easy to implement, and not very demanding regarding computational resources. However, urban areas often present strong $NO_2$ spatial gradients that the official monitoring network cannot correctly characterize due to its low spatial representativeness (Vardoulakis et al., 2005; Santiago et al., 2013; Duyzer et al., 2015a). To overcome this limitation and produce accurate surface $NO_2$ maps, urban or microscale-LUR models rely on Low-Cost Sensors (LCS), typically restricting the temporal coverage to a few weeks. Works dealing with microscale-LUR models have used different types of LCS, including passive dosimeters, which report period-averaged concentrations (Perelló et al., 2021a; Su et al., 2009), time-dependent LCS (Munir et al., 2020; Weissert et al., 2019), or mobile LCS campaigns (Wang et al., 2021). Due to the lack of experimental campaigns monitoring at high spatial and temporal (hourly) resolutions consistently over a whole year, current microscale-LUR studies typically cannot target the hourly averaged $NO_2$ maximum level (200 $\mu g/m^3$) regulated by the 2008 European Ambient Air Quality Directive (AAQD) (2008/EC/50).

Physics-based urban air quality models can generate hourly pollutant concentration estimates, overcoming the temporal limitation of microscale-LUR models. Currently, these systems usually consist of the coupling between a regional chemical transport model, which accounts for the long-range transport of pollutants, and an urban scale dispersion model. The last one can be based on semi-empirical relations such as Gaussian dispersion models and mass exchange global parameterizations (e.g. Soulhac et al. (2017); Kim et al. (2018); Benavides et al. (2019); Denby et al. (2020); Hood et al. (2021), or an obstacle resolving dispersion model using Computational Fluid Dynamics (Kwak et al., 2015; Auvinen et al., 2017). Despite the recent efforts to improve urban dispersion modeling systems, they are afflicted by persistent uncertainties and biases, notably due to the difficulty of prescribing accurate boundary conditions and emissions at the street scale, and reproducing the turbulent phenomena within the urban canopy.

In order to reduce model uncertainties, data-fusion methods can be employed to post-process model outputs and obtain more reliable $NO_2$ exposure maps. Several works have used monitoring station data to build data-fusion methods, either relying solely on urban dispersion models to explain the spatial distribution (Tilloy et al., 2013) or adding different spatial information (e.g. traffic intensity, satellite data, or land use cover) as proxies in addition to the model output (Horálek et al., 2006; Chen et al., 2019; Zhang et al., 2021; Dimakopoulou et al., 2022). In urban areas, the usual low density of monitoring stations has motivated the development of data-fusion methods that integrate LCS campaigns to better explain the spatial distribution of $NO_2$ at the street-scale. For instance, the works of Schneider et al. (2017) and Mijling (2020) combine time-resolved LCS hourly data with an urban model output to improve the $NO_2$ characterization at a high-spatial resolution. Schneider et al. (2017) use a popular geostatistic technique, Universal Kriging, considering the time-aggregated annual mean of an urban model as a *basemap* (or *climatology*) to explain the long-term spatial gradients at the street-scale, while the time-dependent LCS network

explains the short-term temporal behavior. However, the temporal coverage of their results is restricted to a few weeks in which measurements are available. Thus, compromising their use to systematically estimate hourly $NO_2$ exposure levels for extended periods, in the order of years.

By combining model and observational data, advanced data-fusion methods can provide typically unbiased estimates of pollutant concentrations at the street scale. However, another piece of information of crucial importance is the uncertainty of the estimated concentrations, as it can help decision-making or support the design of environmental epidemiological studies (Gryparis et al., 2009). The Universal Kriging methodology provides the error variance of its predictions, which has already been used as a measure of the uncertainty on data-fusion results of $NO_2$ at the street scale (Schneider et al., 2017). However, the validity of the confidence intervals and the normality of error distribution in this application remains to be investigated.

Our study presents a data-fusion methodology considering a microscale-LUR model, in addition to the hourly monitoring data, to bias-correct hourly $NO_2$ estimates of an urban dispersion model at high spatial resolution (20m × 20m). Similarly to Schneider et al. (2017), our work also relies on the basemap concept. However, contrary to previous studies, we have derived it using a microscale-LUR model based on 840 samplers from recent passive dosimeters campaigns (Perelló et al., 2021a; Benavides et al., 2019). Thus, the basemap accounts for the spatial patterns, whereas the temporal behavior is characterized by the hourly urban model output and hourly monitoring data. This approach can be very convenient for applying data-fusion methods in cities where period-averaged LCS campaigns are available but lack time-dependent LCS data throughout the year, which is usually the case. To assess the benefits of considering such microscale-LUR basemap, we compare two different data-fusion methods: (i) Universal Kriging combining hourly observations with the hourly outputs of a street-scale Gaussian dispersion model, namely UK-DM, and (ii) Universal Kriging combining the above items and the microscale-LUR model, namely UK-DM-LUR. The data-fusion methods are applied in Barcelona city (Spain) for the entire year 2019. An original aspect of the present study is the empirical validation of the UK-based uncertainties and their translation into street-scale probabilities of exceedance of the hourly and annual regulatory thresholds.

The paper is structured as follows: the observational data is described in Sect. 2.1. The Gaussian dispersion air quality model CALIOPE-Urban used to produce hourly high-resolution fields of surface $NO_2$ concentrations is described in Sect. 2.2, while the microscale-LUR method is explained in Sect. 2.3. A detailed description of the data-fusion methods is given in Sect. 2.4. Results of the microscale-LUR model are presented in Sect. 3.1, and Sect. 3.2 discusses the results of the data-fusion methodologies. Finally, conclusions and final remarks are provided in Sect. 4.

## 2   Data and methodology

We compare two data-fusion methods (UK-DM and UK-DM-LUR) illustrated in Fig. 1. Below we describe each process and dataset used to derive them.

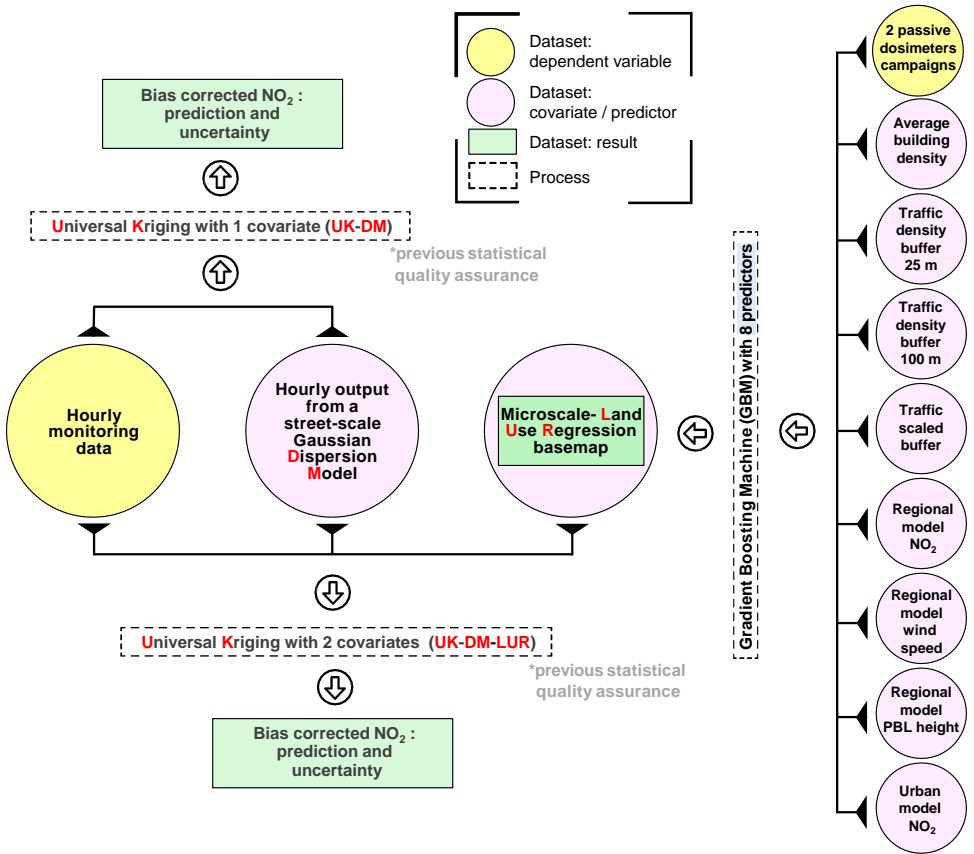

**Figure 1.** Workflow of the two studied data-fusion methodologies. Hourly data from monitoring stations are combined with hourly dispersion model results (UK-DM) and the time-invariant microscale-LUR basemap (UK-DM-LUR). PBL stands for Planetary Boundary Layer.

## 2.1 Study domain and observational NO$_2$ data

Barcelona (Fig. 2) is the second most populated city in Spain and the tenth in Europe, with approximately 1.660.000 inhabitants and 102 km$^2$ ($\sim$ 16.300 people per km$^2$). It is located on the northeast coast of Spain, between the Mediterranean Sea and the

Collserola mountains. The city has a Mediterranean climate characterized by the dominance of sea breeze during the warm season, shallow boundary layer development, and recirculation of air pollutants (Jorba et al., 2004).

Hourly NO$_2$ observational data for 2019 are obtained from the Catalan Atmospheric Pollution Surveillance Network (XVPCA) measurement points in the Barcelona urban and surrounding areas. There are 13 stations available on the Barcelona agglomeration (Fig. 2), with a percentage of availability of hourly data greater than 93 %. *Gràcia* and *Eixample* are urban traffic

monitoring stations, *Segnier*, *Observatori Fabra* and *Jardins* are sub-urban background stations, and the remaining 8 correspond to urban background stations. The *Observatori Fabra* station is not used in our data-fusion methodology since its inclusion significantly degraded the data-fusion skills in the urban environment. This is expected since the station is located

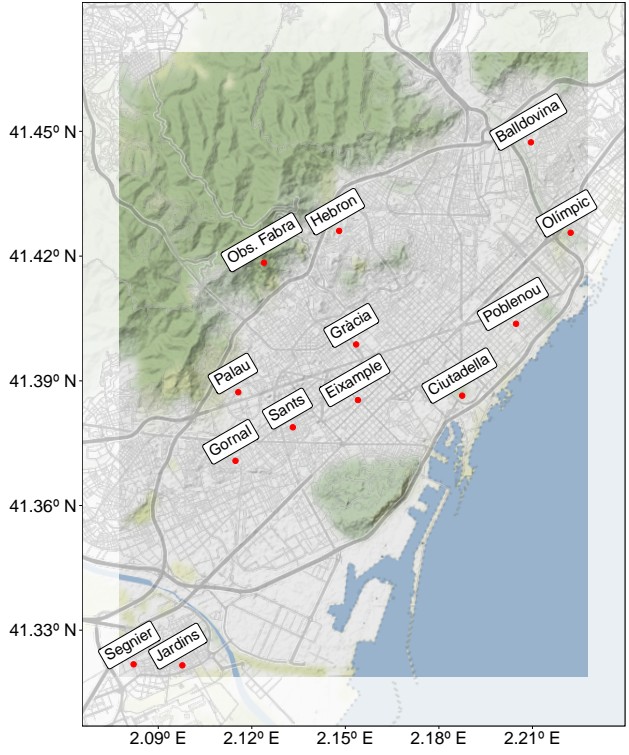

**Figure 2.** Domain of study and location of the referenced monitoring stations. The map has been generated using *ggplot2* (Wickham (2016)) and *ggmap* (Kahle and Wickham (2013)) R packages (R Core Team (2013)), and data from OpenStreetMap. © OpenStreetMap contributors 2017. Distributed under the Open Data Commons Open Database License (ODbL) v1.0. Map tiles are © Stamen Design, under a Creative Commons Attribution (CC BY 3.0) license.

on a hill relatively far from built-up areas. In fact, it is not exactly an urban station because it measures air pollution above the urban canopy while the other stations measure pollution within the urban canopy. We are aware that by removing this station, we may lose relevant information on the low $NO_2$-level regions surrounding the city. However, the main goal of our urban model is to characterize $NO_2$ exceedances in critical trafficked areas. Therefore, we decided to exclude the *Observatori Fabra* station.

Two different $NO_2$ passive dosimeter experimental campaigns (Fig. 3) are considered to derive the microscale-LUR model: the xAire citizen science campaign (Perelló et al., 2021a, b) composed of 725 samplers and deployed between February 16th and March 15th, 2018, and the 2-week measurement campaign of the *Institute of Environmental Assessment and Water Research - Spanish National Research Council* (IDAEA-CSIC), that deployed 175 $NO_2$ samplers across Barcelona during February and March 2017 (Benavides et al., 2019). Both campaigns used Palmes-type $NO_2$ diffusion tubes (Palmes et al. (1976)) to sample the $NO_2$ levels, which implies an estimated uncertainty of $\pm$ 25 %, as reported in Kuklinska et al. (2015).

## 2.2 Street-scale air quality model: CALIOPE-Urban

Hourly high-resolution concentrations of surface $NO_2$ at street-scale over the city of Barcelona are estimated using the CALIOPE-Urban multi-scale air quality model (Benavides et al. (2019)). CALIOPE-Urban accounts for the dispersion of traffic emissions at high spatial resolution using the R-LINE Gaussian dispersion model (Snyder et al. (2013); Venkatram et al. (2013)). As described in more detail in Benavides et al. (2019), R-LINE is adapted to street-canyons by taking into account road-link traffic emissions (Guevara et al. (2020)), meteorological variables (e.g., wind speed and direction, Monin–Obukhov

length and planetary boundary layer height) and buildings morphology (e.g., building density and height, and street orientation). The chemical balance between $NO_x$ and $NO_2$ is computed based on the generic reaction set (Valencia et al. (2018)) assuming clear-sky conditions and uncoupling chemistry from transport phenomena; in other words, the aging of pollutants is solely a function of wind speed and the distance between source and receptors.

At the regional scale, CALIOPE-Urban relies on the regional air quality modeling system CALIOPE (Baldasano Recio et al.

(2011)) for predicting urban background $NO_2$ concentration. The regional CALIOPE accounts for long-range transport of pollutants using three nested domains at increasing resolutions: 12 km × 12 km for the European region, 4 km × 4 km for the Iberian Peninsula, and 1 km × 1 km for the region of Catalonia (Baldasano Recio et al. (2011), Pay et al. (2014)). The urban background $NO_2$ concentrations obtained with regional CALIOPE are combined with the R-LINE dispersion results using a dedicated parameterization of the vertical mixing (Benavides et al. (2019)).

In this work, CALIOPE-Urban employs a non-uniform mesh refined at the edge of traffic roads and coarser in low-gradient regions of $NO_2$. This type of mesh accelerates the calculations and reduces memory demand. The refined grid zones have a resolution of 25 m × 25 m, progressively degrading to 500 m × 500 m in the regions of low $NO_2$ gradients. To facilitate their visualization, these $NO_2$ concentrations are finally interpolated over a uniform mesh with a resolution of 20 m × 20 m. CALIOPE-Urban has been evaluated and successfully used in the framework of several impact studies, including the works of

Benavides et al. (2021) and Rodriguez-Rey et al. (2022).

## 2.3 Microscale-LUR model using Gradient Boosting Machine (GBM)

A non-linear microscale-LUR model based on passive dosimeter campaigns is used to produce an observation-based climatological view of the $NO_2$ concentrations at a high spatial resolution over Barcelona city. While the monitoring stations and the urban dispersion model provide information on the pollutants' short-term temporal behavior, the microscale-LUR basemap

(long-term mean) remains constant in time. Its main goal is to provide reliable long-term spatial variability patterns of $NO_2$ at high resolution using observational data and other urban information.

The target variable of the microscale-LUR model is the time-averaged concentrations of the two different $NO_2$ experimental campaigns described in Sect. 2.1 and represented in Fig. 3. We have discarded the xAire samplers related to playgrounds and classrooms, so we are using the remaining 669. In order to combine the xAire and IDAEA-CSIC campaigns, we have

annualized both following the procedure described in Perelló et al. (2021b): for each station, an adjustment factor is computed as the ratio between the observed 2017 annual mean and the average over the period of the experimental campaign. Then, the

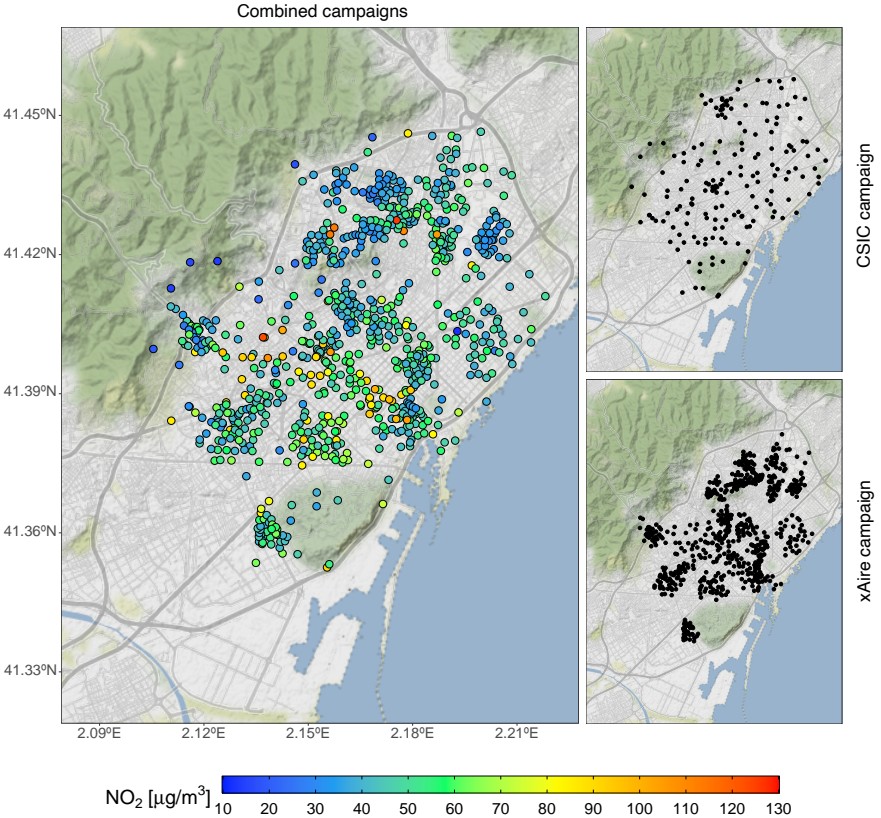

**Figure 3.** Sampler locations of the two different NO$_2$ experimental campaigns used to train the microscale-LUR model. The left panel shows the NO$_2$ values and the locations of the combined campaigns. The top- and bottom-right panels show the CSIC and xAire campaign locations, respectively. The colour scale refers to the 2017 annualized NO$_2$ values, in $\mu g/m^3$. The map has been generated using *ggplot2* (Wickham (2016)) and *ggmap* (Kahle and Wickham (2013)) R packages (R Core Team (2013)), and data from OpenStreetMap. © OpenStreetMap contributors 2017. Distributed under the Open Data Commons Open Database License (ODbL) v1.0. Map tiles are © Stamen Design, under a Creative Commons Attribution (CC BY 3.0) license.

average of this factor over all stations is used to scale all passive samplers to the 2017 annual mean. This scaling assumes that the ratio does not depend on location and can be applied to all samplers. Despite adding some noise to the experimental results, it corrects the bias induced by environmental conditions (e.g., wind speed, atmospheric stability, precipitation, radiation, temperature) and also allows combining both campaigns, producing a dataset of 844 samplers on which the microscale-LUR model relies. Note that the microscale-LUR model is trained using experimental campaigns deployed in February and March. As a result, even though the annualization process corrects the NO$_2$ levels and the predictors are expressed as annual averages, the captured spatial gradients may still have a significant seasonal bias.

The potential predictors of the microscale-LUR model are shown in Table 1. The geometric variables are calculated from the *Institut Cartogràfic i Geològic de Catalunya* (ICGC) and *Plan Nacional de Ortografía Aérea* (PNOA (2020)). Traffic-related predictors consist of traffic density ($t$) for different circular buffer sizes. Being $a$ the radius of the buffer, $t_a$ is computed following Eq. (1) expressed in [*vehicles · m / s*]:

$$t_a = \sum_{i=1}^{n} AADT_i \cdot l_{a,i}, \tag{1}$$

where $i$ represents the street segment, $n$ is the number of street segments over the circular area of $\pi a^2$, $l_i$ is the length of the street segment $i$ within the buffer, and $AADT_s$ is the average daily traffic of the street segment $s$ expressed in vehicles per second. The $t_a$ predictors associated with the smaller buffers (5, 10 and 15 m) have highly skewed distributions, given that most values across the map are null. To avoid training the microscale-LUR with skewed predictors, we introduce here the *traffic scaled* variable, $s$, which combines all buffers as follows:

$$s = \frac{1}{N} \sum_{a} \frac{t_a}{\pi a^2}, \tag{2}$$

where $N$ is the number of buffers (12 in our case). Traffic data are extracted from the road-link traffic network of the HERMESv3 bottom-up emission model (Guevara et al. (2020)). We also considered $NO_2$, Planetary Boundary Layer height and wind speed annual means from the regional air quality modeling system CALIOPE as potential predictors, together with the $NO_2$ annual mean from the air quality model CALIOPE-Urban.

A recursive feature elimination method has been applied to remove highly correlated or uninformative features. We have used the simple backward selection algorithm implemented in the R package (Kuhn, 2008), which starts with the full-featured model and gradually removes the least important feature while monitoring the *RMSE* in Cross-Validation (CV). The goal is to obtain the simplest model with the lowest *RMSE* to gain generalization and interpretability. The final microscale-LUR model includes the following eight predictors: average building density; traffic buffers of 25, 100 m, and the traffic scaled variable; all the annually averaged data from the regional CALIOPE modeling system; and the annual average $NO_2$ from CALIOPE-Urban.

To account for non-linear relations among the predictors and the target variable, we used the Gradient Boosting Machine (GBM) algorithm implemented in the *R* package *gbm* (Ridgeway, 2004). GBM is a popular machine learning algorithm (Natekin and Knoll, 2013) that has shown excellent results in terms of accuracy and generalization when compared to other learning algorithms (Caruana and Niculescu-Mizil (2006)). The GBM hyper-parameters (shrinkage rate, interaction depth, minimum observation per node, and bag fraction) are optimized based on the minimum mean cross-validated error and a grid search algorithm. Additionally, following the work of Chen et al. (2019), we exploit the potential spatial correlation of the GBM residuals by interpolating and adding them to the predicted values. The interpolation is done with an Ordinary Kriging (OK) (Wackernagel, 2003).

One could think that skipping over the LUR computation by directly using all its time-invariant information (passive dosimeter campaigns, urban geometry, traffic-related data, and annual-averaged model results) as covariates in the Universal Kriging

| Type | Num. | Variable | Resolution |
|---|---|---|---|
| Urban geometric | 1 | Average building density | Square buffer of 250m × 250m |
| | 2 | Average building height | |
| | 3 | Maximum building height | |
| | 4 | Standard deviation building height | |
| Traffic-related | 5-16 | Simulated vehicular traffic densities | Circular buffers of 5, 10, 15, 25, 50, 100, 300, 500, 1000, 2000, 3000 and 4000 m of radius |
| | 17 | Traffic scaled | Linear combination of the buffers above |
| Output from the regional modeling system CALIOPE (lowest layer) | 18 | $NO_2$ | Uniform mesh of 1km × 1km |
| | 19 | Planetary boundary layer height | |
| | 20 | Wind speed | |
| Output from the CALIOPE-Urban model | 21 | $NO_2$ | Non uniform mesh (25m × 25m to 500m × 500m) |

**Table 1.** The microscale-LUR model contemplates the use of these 21 potential predictors.

would simplify the workflow. However, there are two main drawbacks to doing so. On the one hand, in contrast to the GBM, the Universal Kriging assumes linear relations between covariates and the observed $NO_2$, which is not necessarily true for this case. On the other hand, when considering a large number of covariates with only 12 monitoring stations, strong spurious correlations lacking physical meaning are prone to happen, wrongly driving the final solution (Hengl et al. (2007)). Thus, gathering all static information in the single LUR covariate offers more robust results while permitting the addition of predictors using non-linear regression models.

### 2.4 Universal Kriging as a data fusion methodology for spatial bias correction

Microscale-LUR model and the hourly CALIOPE-Urban outputs are combined with observational $NO_2$ data from the monitoring stations using the geostatistical technique Universal Kriging, which is commonly used for spatial interpolation. This methodology predicts a random variable $Z$, in a target point $\mathbf{x}$, based on a combination between a multi-linear regression analysis with external variables $f$, referred as *covariates*, and a pure spatial interpolation considering the auto-correlation structure of the regression residuals. In our case, the variable $Z$ corresponds to the monitoring data while the covariates are CALIOPE-Urban and our microscale-LUR model. A simple (multi)linear regression model is convenient here, given the low number (12) of available monitoring stations within the computational domain. Universal Kriging assumes the following relation (Cressie, 1993):

$$Z(\mathbf{x}) = \sum_{l=0}^{L} a_l f_l(\mathbf{x}) + R(\mathbf{x}), \tag{3}$$

where L equals 1 in the UK-DM approach and 2 in the UK-DM-LUR; $a_l$ are the non-zero coefficients from the multi-linear regression between the observations and the covariates $f_l$, with $f_0(\mathbf{x}) = 1$ by convention; and $R(\mathbf{x})$ is the residual random field. The deterministic part of the variable $Z$ is explained by a linear combination of the covariates, while the residual random field is considered to have zero mean and to be spatially auto-correlated. The main advantage of this method is that depending on the strength of the correlation between covariates and observations, Universal Kriging gives more weight either to the multi-linear regression or to the spatial interpolation of the residuals (Hengl (2009)). Thus, providing a robust data-fusion method that adapts to the quality of the model output.

As a Gaussian process, Universal Kriging estimates the variance of its predictions ($\sigma^2$) coming from both, the multi-linear regression ($\sigma^2_{MLR}$), and the spatial interpolation ($\sigma^2_{SI}$) steps, as

$$\sigma^2(\mathbf{x}) = \underbrace{\sum_{\alpha=1}^{m} w_\alpha \cdot \gamma_R(\mathbf{x}_\alpha - \mathbf{x})}_{\sigma^2_{SI}} + \underbrace{\sum_{l=0}^{L} \lambda_l f_l(\mathbf{x})}_{\sigma^2_{MLR}}, \tag{4}$$

where $m$ is the number of monitoring stations; $w_\alpha$ are the spatial interpolation weights associated with each measurement point; $\lambda_l$ are the $L+1$ Lagrange multipliers used to minimize the variance error; and $\gamma_R(\mathbf{x}_\alpha - \mathbf{x})$ stands for the variogram, which characterizes the spatial structure of the residuals (Chiles and Delfiner (1999)). Thus, the variance of a prediction reflects how far the unmeasured location is from the observation points, as well as from the feature space in which the regression model has been calibrated, i.e., the extrapolation effect (Hengl (2009)). Our Universal Kriging implementation relies on the $R$ package *gstat* (Pebesma (2004), Gräler et al. (2016)).

To normalize the distribution of the NO$_2$ data and to ensure positive predicted values, we have applied the Universal Kriging described above after transforming NO$_2$ data into the log-space. However, results need to be back-transformed to the original scale. Following the work of Cressie (1993), the back-transformation is performed as

$$\hat{Z}(\mathbf{x}) = exp(\, Z_l(\mathbf{x}) + \sigma^2_l(\mathbf{x})/2 \,), \tag{5}$$

$$\hat{\sigma}^2(\mathbf{x}) = (exp(\sigma^2_l(\mathbf{x})) - 1) \cdot exp(2 \cdot Z_l(\mathbf{x}) + \sigma^2_l(\mathbf{x})), \tag{6}$$

where $\hat{Z}(\mathbf{x})$ and $\hat{\sigma}^2(\mathbf{x})$ respectively represent back-transformed prediction and variance at the target point, while $Z_l(\mathbf{x})$ and $\sigma^2_l(\mathbf{x})$ are prediction and variance in log-space, respectively.

Assuming a normal distribution of the error, the probability of exceedance ($\mathcal{P}$) of a certain limit value ($\mathcal{L}$) can be computed as

$$\mathcal{P}(\mathbf{x}) = 1 - F\left( \frac{\mathcal{L} - \hat{Z}(\mathbf{x})}{\hat{\sigma}(\mathbf{x})} \right), \tag{7}$$

where $F$ is the normal cumulative distribution function.

### 2.4.1 Statistical metrics to evaluation data-fusion skills

225 Statistical performance is assessed by Leave-One-Out-Cross-Validation (LOOCV), which consists of performing the data-fusion considering all monitoring stations except one kept to cross-validate the results. For each LOOCV we present the *Coefficient of Efficiency* (*COE*), the *Root Mean Square Error* (*RMSE*), the *Mean Bias (MB)*, and the *Correlation Coefficient* (*r*) defined as:

$$COE = 1 - \frac{\sum_{i=1}^{k}|M_i - O_i|}{\sum_{i=1}^{k}|O_i - O|} \tag{8}$$

230
$$MB = \frac{1}{k}\sum_{i=1}^{k} M_i - O_i \tag{9}$$

$$r = \frac{1}{k-1}\sum_{i=1}^{k}\left(\frac{M_i - M}{\sigma_M}\right)\left(\frac{O_i - O}{\sigma_O}\right) \tag{10}$$

$$RMSE = \sqrt{\frac{1}{k}\sum_{i=1}^{k}(M_i - O_i)^2} \tag{11}$$

where $k$ is the total number of observations; $O_i$ and $M_i$ are the observed and modelled $i$ values, respectively; $O$ and $M$ are their respective means; and $\sigma_O$ and $\sigma_M$ refer to their standard deviation.

### 235 2.4.2 Spatial auto-correlation structure of NO$_2$ levels

In the Universal Kriging context, the variogram describes the spatial auto-correlation structure of the residual random field. In our case, the limited number of monitoring stations makes extracting a meaningful spatial structure challenging. For this reason, we estimate the residual variogram based on the dosimeters campaigns. This decision, however, entails a substantial limitation due to the assumption of a static variogram. We rely only on the IDAEA-CSIC campaign (discarding the xAire

240 campaign for the variogram derivation) to avoid extra assumptions on combining campaigns. Additionally, we considered an isotropic variogram. All these assumptions impact the variance error estimated by the Universal Kriging (Brus and Heuvelink (2007)). To assess the impact of such assumptions, an analysis of the estimated variance in LOOCV is carried out in Sect. 3.2.

The variogram is fitted using the Matérn model with the Stein's parametrization implemented in the *automap* package (Hiemstra et al., 2008) setting the smoothing parameter $\kappa = 0.2$. The resulting variogram model is characterized by a $5 \times 10^{-2}$

245 *partial sill*, $3 \times 10^{-5}$ *nugget* and a *range* of 620 m. Following the work of Denby et al. (2007), we have optimized the *range* value to minimize the RMSE of the Universal Kriging. The *range* estimates the distance at which the data are no longer correlated. To optimize it, we performed a CV at all monitoring stations varying the range from 1 to 10 km every 1 km, keeping all other model parameters constant. We obtained the best results for the *range* of 5 km, improving the *r* by 4 %, the *COE* by 14 %, and the *RMSE* by -9 % with respect to the Universal Kriging using the original range value of 620 m.

### 2.4.3 Statistical quality assurance of the (multi)linear regression

The correlation coefficient ($r$) and the regression coefficient (slope) of the regression model between covariates (CALIOPE-Urban and the microscale-LUR model) and observations are checked before including covariates in the Universal Kriging workflow as indicated in Fig. 1. If a covariate shows a low correlation (p-value > 0.05) with the observations at a specific hour, it is not considered in the regression model, as in the works of Zhang et al. (2021) and Oh et al. (2021). Additionally, if none of the covariates show a significant correlation, we use both covariates to build the regression model. However, to avoid nonphysical hourly maps, the covariates are used only if their regression coefficient is positive, as suggested by Denby et al. (2007). In case all regression coefficients are negative, or there are less than 4 observations available in a specific hour, the Universal Kriging is not performed and the results of the data-fusion method are directly the raw dispersion-model output. Following the above criteria, the percentage of cases with fewer than 4 monitoring observations is relatively small, 0.034 % (3 hours), and is the same for each kriging application. For the UK-DM methodology, 14.11 % of the hours have not been corrected due to negative regression coefficients. On the other hand, for the case of UK-DM-LUR, only 1.47 % of the hours have been discarded due to a negative regression coefficient in both covariates. As Benavides et al. (2019) identified, the poor skills of the urban model are attributed to low wind speeds and atmospheric stability situations, for which the performance of the mesoscale model decreases. Concerning the static microscale-LUR basemap, the poor correlation on an hourly basis is associated with hours that significantly deviate from the average behavior.

## 3 Results and Discussion

Results are organized into two sections. Firstly, in Sect. 3.1, we estimate the microscale-LUR model performance and present the obtained $NO_2$ basemap. Secondly, in Sect. 3.2, the data fusion methodologies are discussed in terms of statistical performance, uncertainty quantification, and exceedance probability maps. All the maps presented in this section have been generated using *ggplot2* (Wickham (2016)) and *ggmap* (Kahle and Wickham (2013)) R packages (R Core Team (2013)), and data from OpenData BCN (Ajuntament de Barcelona (2019)) and OpenStreetMap. © OpenStreetMap contributors 2017. Distributed under the Open Data Commons Open Database License (ODbL) v1.0. Map tiles are © Stamen Design, under a Creative Commons Attribution (CC BY 3.0) license.

### 3.1 Microscale-LUR model

#### 3.1.1 Performance assessment

The GBM-based microscale-LUR model is evaluated using two nested K-fold CV, the inner one for tuning the model (*Training-validation set*) and the outer one for testing the model on different parts of the dataset (*Test set*). Such a procedure aims at giving a reliable estimate of the expected performance. We use an outer 10-fold CV and an inner 4-fold CV as illustrated in Fig. 4. The tuning of the model is performed through a grid search over the following hyperparameters: shrinkage rate (with values

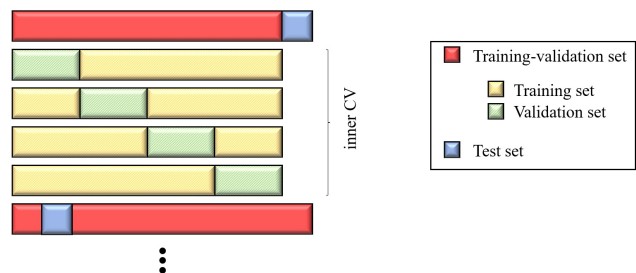

**Figure 4.** Scheme of the outer 10-fold CV and the inner 4-fold CV applied for the GBM training.

| Model | | n | COE | MB ($\mu g/m^3$) | r | RMSE ($\mu g/m^3$) |
|---|---|---|---|---|---|---|
| Microscale-LUR | Training-validation set | 7600 | 0.30 | 0.15 | 0.69 | 11.38 |
| | Test set without adding the residuals | 840 | 0.24 | 0.22 | 0.62 | 12.17 |
| | Test set adding the residuals | 840 | 0.27 | -0.27 | 0.64 | 11.87 |
| Raw CALIOPE-Urban | Annual mean | 840 | 0.13 | -0.81 | 0.54 | 13.68 |

**Table 2.** Statistical results of the microscale-LUR model in nested CV. The 2017 annual mean concentration of $NO_2$ of the raw dispersion model (CALIOPE-Urban) is also shown. The parameter *n* stands for the number of data points used to compute the statistics.

ranging from 0.001 to 0.05 every 0.001), the interaction depth (from 1 to 4 every 1), the minimum observation in a node (from 5 to 15 every 1), and the bag fraction (0.5 and 0.65).

Results are given in Table 2, together with the performance reference of the annual mean $NO_2$ concentration obtained directly from CALIOPE-Urban. As explained in Sect. 2.3, we exploit the spatial auto-correlation of the LUR residuals to improve its estimation. To do so, the LUR residuals at the training locations are interpolated at the test locations by applying an OK. Then,

they are added to the predictions to obtain the corrected results (*Test set adding the residuals* in Table 2).

Table 2 shows that the LUR model significantly improves the CALIOPE-Urban results. Also, the addition of the residuals slightly increases the statistical performance. The *Training-validation set* results are not perfectly fitted and are only slightly better than the *Test set* results, indicating that the LUR is not overfitted and has good capabilities to predict unseen data.

We show in Fig. 5 the scatter plots of the *annual mean CALIOPE-Urban* and the *Test dataset* results with and without adding

the residuals, along with the observational uncertainty ranges indicated by the dashed red lines ($\pm 25$ % according to Kuklinska et al. (2015)). Although a large portion of the predicted values for the LUR model with the residual correction lies within the uncertainty range, difficulty in predicting values of $NO_2$ higher than 80 $\mu g/m^3$ can be observed. We attribute this behavior to the limited number of points in this range, which can weaken the model training, particularly in the nested CV context, but also to the already poor predictive skills of CALIOPE-Urban in this concentration range as seen in Fig. 5a.

Comparing these results with previous works, the resulting correlation coefficient ($r$) is lower than a LUR model fitted with the xAire campaign data ($r = 0.74$ in LOOCV) reported in Perelló et al. (2021a). However, Perelló et al. (2021a) used only 370

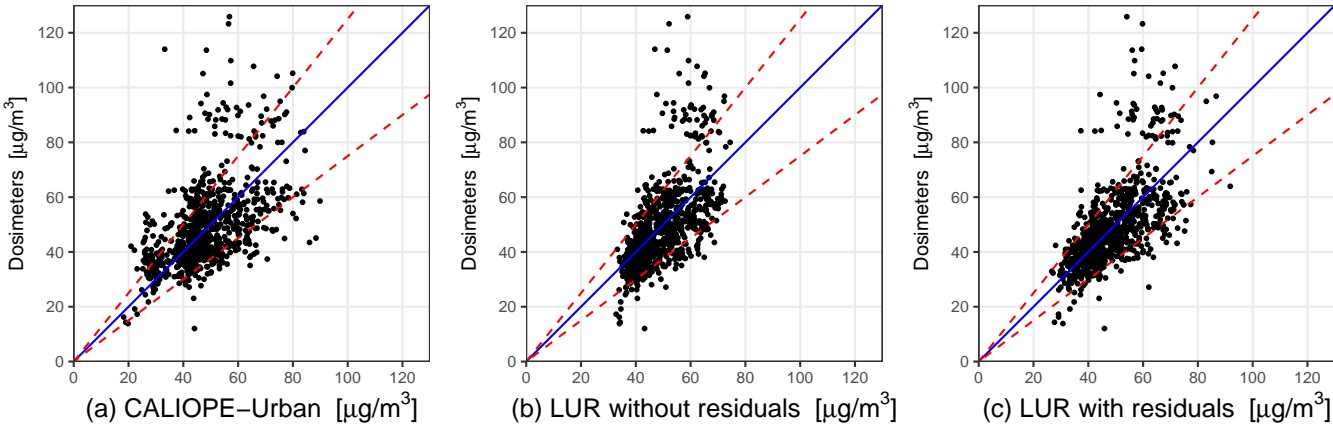

**Figure 5.** (a) Raw annual mean of CALIOPE-Urban NO$_2$ concentrations, (b) microscale-LUR model results, without the interpolated residuals, and (c) microscale-LUR model results, with the interpolated residuals, versus the annualized passive dosimeters campaigns. These figures use the test sets in which the performance of the microscale-LUR model has been assessed. The red dashed lines report passive dosimeter uncertainty ($\pm 25\%$) and the identity line is represented in blue. The statistical results are shown in Table 2.

outdoor sampling sites out of the 669 available. They excluded samplers close to traffic and street intersections, achieving a skilled urban LUR model. Even if the correlation coefficient is slightly lowered, we have considered all outdoor sampling sites (along with the IDAEA-CSIC campaign) to capture as much as possible the NO$_2$ spatial trends. On the other hand, the work
of Munir et al. (2020) reported a microscale-LUR model based on 40 time-dependent LCS with slightly lower performance in CV than the present one ($r = 0.56$). Moreover, Munir et al. (2020) also reported a $r$ value of 0.53 when the non-linear LUR model is based on the combination of 188 period-averaged and 40 time-dependent LCS. A key aspect of the present data-fusion methodology is that the microscale-LUR model results better explain the annualized passive dosimeters campaigns compared to the reference CALIOPE-Urban annual mean. Therefore, they are subsequently considered as a covariate in the Universal
Kriging methodology, as further explained in the next Sect. 3.1.2. An assessment regarding the necessary amount of samplers to derive a robust microscale-LUR model is presented in Appendix A.

### 3.1.2 Microscale-LUR basemap

We proceed to train the microscale-LUR model with the residual correction using all available sampling sites. Figure 6 compares the long-term NO$_2$ patterns from the microscale-LUR basemap (Fig. 6a) with the NO$_2$ 2019 annual mean of CALIOPE-
Urban (Fig. 6b). Notice that the goal of the basemap is to correct the long-term spatial variability of NO$_2$. Thus, Fig. 6 highlights the differences in spatial patterns rather than differences in absolute NO$_2$ values. The resulting basemap shows a qualitatively consistent NO$_2$ distribution: the major trafficked roads of the city and the port area are the most polluted locations, while the Collserola mountains and the sea bordering the city have moderate NO$_2$ levels. Although both figures show similar NO$_2$ patterns, local differences from experimental information can be observed in Fig. 6a. For instance, there is a noticeable increase in

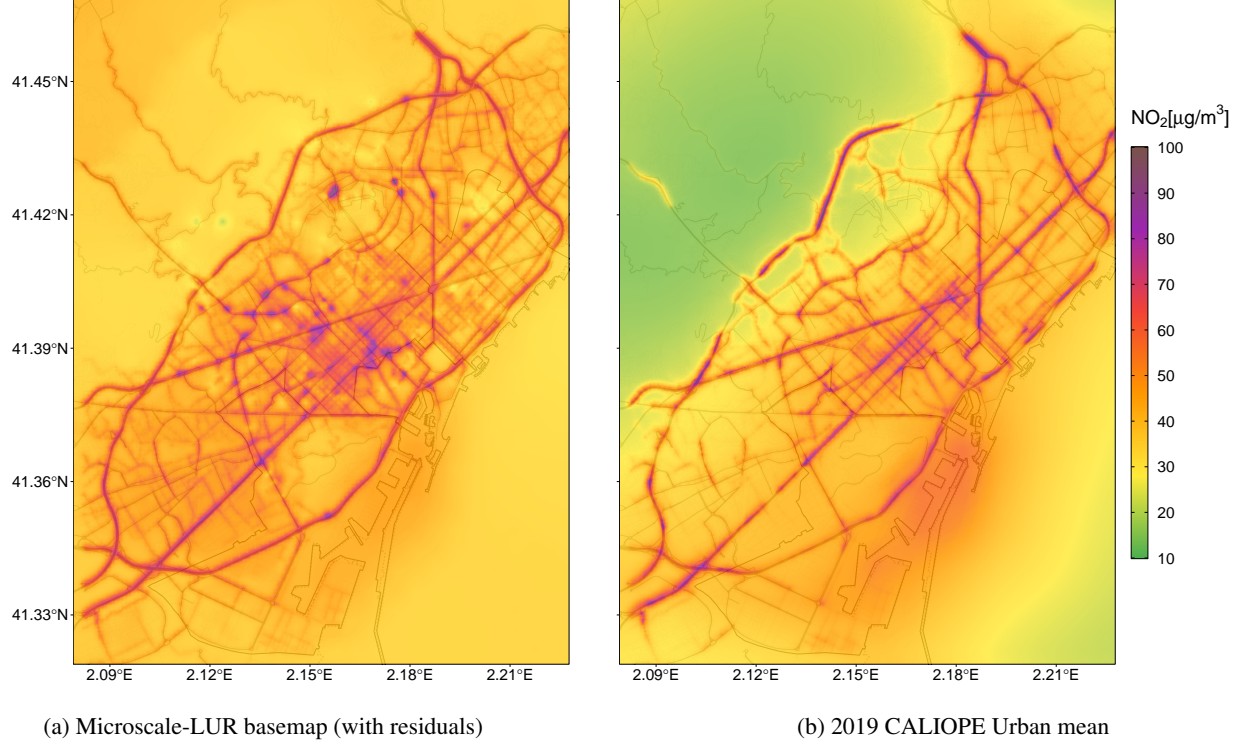

(a) Microscale-LUR basemap (with residuals)    (b) 2019 CALIOPE Urban mean

**Figure 6.** (a) Resulting microscale-LUR basemap using all available sampling sites and adding the interpolated residuals, and (b) 2019 annual mean concentration of $NO_2$ of the raw dispersion model CALIOPE-Urban.

$NO_2$ levels for the microscale-LUR basemap (Fig. 6a) in the mountainous north-western area of the study domain. This artifact is probably caused by the spatial distribution of the passive dosimeters campaigns (Fig. 3), which poorly cover this region. The $NO_2$ overprediction of this area is not reflected in the statistical evaluation of the data-fusion since we deliberately omitted the monitoring station located in this area. We excluded this station to improve the data-fusion model's ability to capture $NO_2$ exceedances in built-up areas, which is the main goal of the urban model. As further shown in the statistical results, consid-
ering extensive passive dosimeters information through the microscale-LUR model avoids relying only on the urban model to describe the $NO_2$ gradients and significantly improves the data-fusion methodology.

  The influence of each predictor in the final microscale-LUR model has been computed based on the methodology proposed by Friedman (2001) and implemented in the *R* package *gbm* (Ridgeway (2004)), in which the relative importance of each variable is associated to the reduction of the GBM cost function. Given the chosen set of predictors, the most influential
variable is the $NO_2$ CALIOPE-Urban annual mean with a relative importance of 25.1 %, followed by 17.7 % for the traffic scaled variable and 15.7 % for the average building density. The other predictors exhibited a relative influence under the 15 %, with the $NO_2$ CALIOPE regional mean as the lowest one with 4.3 %.

## 3.2 Data-fusion methodologies

### 3.2.1 Statistical evaluation

In order to quantify the added value of including the microscale-LUR basemap in the data-fusion methodology, two different post-processes (see Fig.1) have been carried out. First, the output of the urban dispersion model CALIOPE-Urban is merged with the monitoring data using Universal Kriging, named UK-DM. Second, the microscale-LUR basemap is added as a covariate in the Universal Kriging workflow, named UK-DM-LUR.

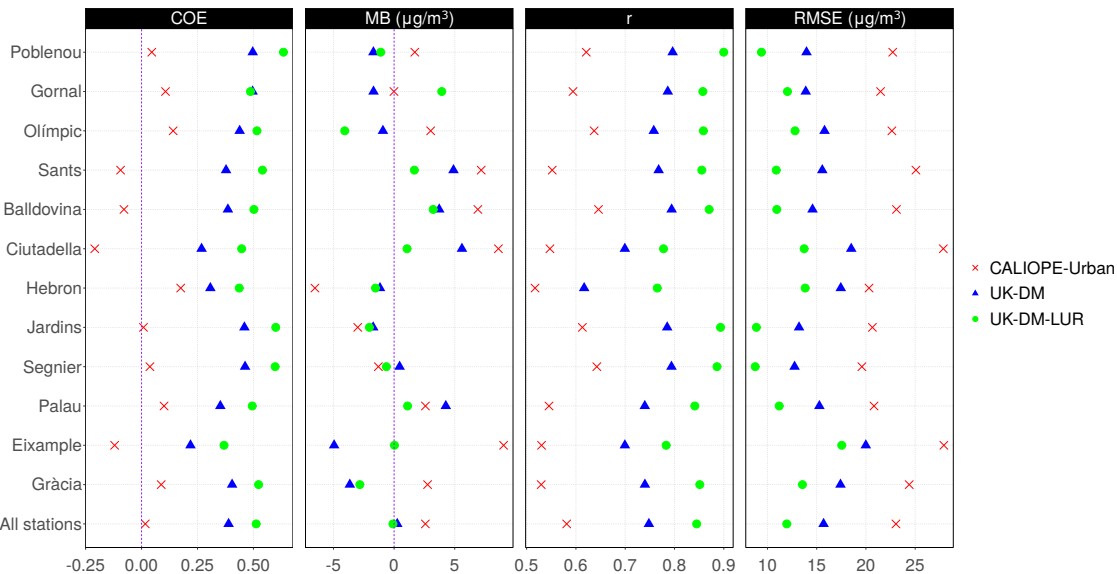

**Figure 7.** Statistical results for each station after applying UK-DM and UK-DM-LUR to 2019 hourly data in LOOCV. In addition, we show the statistical results for the CALIOPE-Urban estimates at each station. The *All stations* row refers to the average over all stations.

Hourly statistical results for the raw CALIOPE-Urban, UK-DM and UK-DM-LUR are shown in Fig. 7 for each monitoring station using all available data of 2019. UK-DM and UK-DM-LUR results have been computed in LOOCV as explained in Sect. 2.4. *Gràcia* and *Eixample* are the urban traffic monitoring stations, and the last row in Fig. 7 corresponds to the average results over all stations. This figure shows that the hourly scale post-processes consistently improve all studied statistical metrics at all monitoring stations, regardless of the monitoring station type. Moreover, adding the LUR basemap as a covariate (UK-DM-LUR) further improves the spatial correction at all stations and for all statistical metrics, except for the *MB* which does not have a clear trend. A negative value of the *COE* reflects a poor predictive capacity, so we highlight that both data-fusion methods achieve a positive *COE* at all considered stations. Almost all stations show a positive *MB* for CALIOPE-Urban indicating a general overestimation of the model, while UK-DM and UK-DM-LUR present almost null *MB* averaged over all stations. The overestimation of CALIOPE-Urban in the monitoring stations may seem contradictory with the negative

bias presented in Table 2 for the passive dosimeters campaigns. However, this could indicate that the highest NO$_2$ values in
Barcelona city are not routinely monitored, as already pointed out in the work of Duyzer et al. (2015b). Regarding the *RMSE*,
the averaged reduction between CALIOPE-Urban and UK-DM is about 32 %, and 24 % between UK-DM and UK-DM-LUR.
For *r*, an averaged improvement of 29 % between CALIOPE-Urban and UK-DM is observed, while the improvement is of 13
% between UK-DM and UK-DM-LUR.

### 3.2.2 Uncertainty quantification

The uncertainty of the Universal Kriging predictions is estimated from the (multi)linear regression and the spatial interpolation
variances, as formulated in Sect. 2.4. The spatial interpolation is based on the variogram, which has been modelled from a
period-averaged passive dosimeter campaign; thus, assuming a static behavior as pointed out in Sect. 2.4.2. Additionally, the
variogram is considered isotropic for simplicity, while we know that in the urban scale, the NO$_2$ autocorrelation structure may
significantly vary depending on the direction with respect to traffic road-links. These assumptions directly impact the error
variance estimated by the Universal Kriging, $\hat{\sigma}^2$. Considering that the interpolation error is normally distributed (N$_{ref}$), the
observation at a specific monitoring station when performing a LOOCV should be within $\pm\hat{\sigma}$ of the predicted value 68 %
of the times, while 95 % and 99.7 % respectively for $\pm 2\hat{\sigma}$ and $\pm 3\hat{\sigma}$, as indicated in Table 3. To assess the normality of
these distributions, Table 3 reports an empirical validation of the percentage of observations falling within the corresponding
error range, again computed in LOOCV. These percentages show that uncertainty is underpredicted for both methods, being
UK-DM-LUR overconfident results slightly better than the UK-DM ones.

|  | $\pm 1\hat{\sigma}$ | $\pm 2\hat{\sigma}$ | $\pm 3\hat{\sigma}$ |
|---|---|---|---|
| N$_{ref}$ | 68 % | 95 % | 99,7 % |
| UK-DM | 47.9 % | 78.0 % | 91.3 % |
| UK-DM-LUR | 51.2 % | 81.3 % | 92.9 % |

**Table 3.** Percentages of observations falling in the $\pm\hat{\sigma}$, $\pm 2\hat{\sigma}$, $\pm 3\hat{\sigma}$ confidence intervals using all stations in LOOCV during 2019. Confidence
intervals are computed based on the hourly predicted values and their standard deviation.

To better understand the behavior of uncertainty estimates, we show in Fig. 8 the probability density functions (PDFs) of
the hourly bias, normalized by the error standard deviation $\hat{\sigma}$, for UK-DM and UK-DM-LUR using all studied monitoring
stations in LOOCV over all available hours in 2019. The error PDFs have normal trends with a slightly negative skew and are
overconfident in accordance with Table 3. Both methodologies, especially UK-DM, exhibit negative skewness. This is because
the corrected model struggles to capture the infrequent high pollution peaks, tending to underestimate them significantly. Thus,
negative biases ($M_h < O_h$) are rare but stronger. On the other hand, the model tends to overpredict moderate observed values
slightly. Therefore, positive biases ($M_h > O_h$) are more frequent and less severe. In agreement with the overall null bias, the
rare strong underestimations are compensated by frequent moderate overestimations.

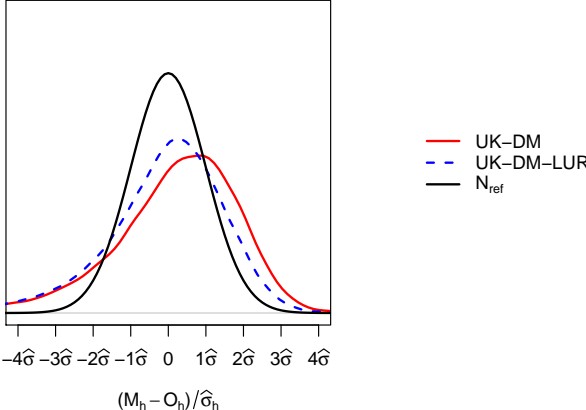

**Figure 8.** PDF of the hourly bias, normalized by the Universal Kriging standard deviation, for all monitoring stations in LOOCV during 2019. The PDFs correspond to the reference Normal distribution ($N_{ref}$), UK-DM and UK-DM-LUR hourly results.

In addition, in Fig. 9, the PDFs are computed by splitting the observed $NO_2$ concentration levels in three different ranges: lesser than 40, greater than 100, and between 40 and 100 $\mu g/m^3$. These PDFs allow us to study the behavior of the error distribution for different $NO_2$ values. This figure shows that larger concentration levels tend to be underestimated, while the smaller ones are overestimated. In all ranges and for both methodologies, the normal trends of the error PDFs are conserved, being the intermediate ranges the closest to the theoretical normal distribution.

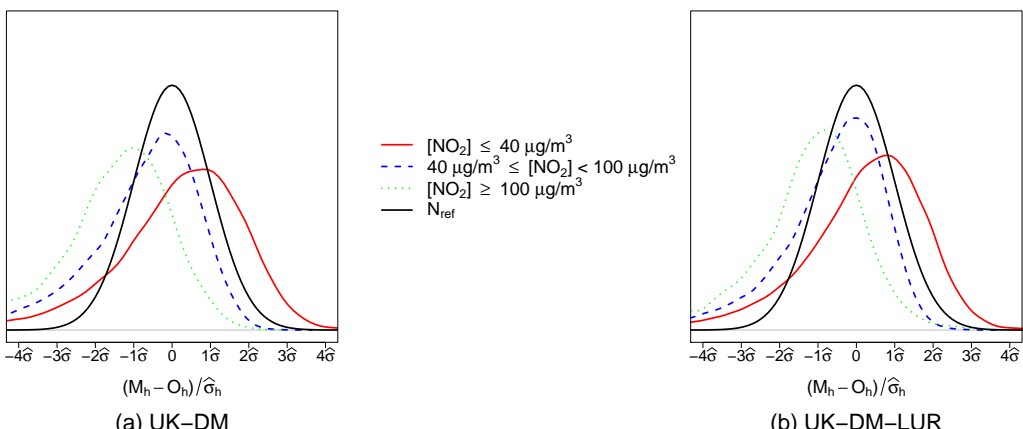

**Figure 9.** PDF by observed $NO_2$ ranges of the hourly bias, normalized by the standard deviation error, for all monitoring stations in LOOCV during 2019 for (a) UK-DM and (b) UK-DM-LUR applications.

### 3.2.3 Street-scale maps

We first analyze the annual mean concentration levels of $NO_2$. Table 4 presents the evaluation in LOOCV of the results post-processed by the UK-DM and the UK-DM-LUR methodologies applied directly to the 2019 annual mean. The presented statistics are computed using a single $NO_2$ averaged value for each station. The annual-based statistics are similar to the hourly results shown in Fig. 7; however, there is a substantial drop in the *RMSE* associated with the bias compensation when averaging the hourly data.

| | COE | MB ($\mu g/m^3$) | r | RMSE ($\mu g/m^3$) |
|---|---|---|---|---|
| UK-DM | 0.25 | 0.20 | 0.74 | 3.93 |
| UK-DM-LUR | 0.38 | 0.37 | 0.83 | 3.24 |

**Table 4.** Statistical results using the 12 monitoring stations after applying UK-DM and UK-DM-LUR directly to the annual mean in LOOCV.

Fig. 10 presents the $NO_2$ annual mean, their associated relative uncertainty, and probability maps of exceeding the 40 $\mu g/m^3$ $NO_2$ annual limit value (AAQD 2008/EC/50) for both, UK-DM and UK-DM-LUR methodologies. The annual mean levels combining the raw model with the monitoring stations data (UK-DM) (Fig. 10a) have similar trends to the raw CALIOPE-Urban (Fig. 6b); however, pollution levels are significantly reduced. Adding the passive dosimeters information through the microscale-LUR basemap (UK-DM-LUR), in Fig. 10d, slightly increases $NO_2$ concentrations, particularly in the city center 385 and secondary roads, where the microscale-LUR basemap (Fig. 6a) exhibits steeper $NO_2$ gradients than CALIOPE-Urban (Fig. 6b).

As expected, the areas surrounding the monitoring stations (presented in Fig.2) show lower relative uncertainty, as can be seen in Figs. 10b and 10e. The higher uncertainty regions, on the other hand, correspond to areas far from monitoring sites and with extreme concentration levels which causes an extrapolation effect in the regression model. When comparing the 390 two uncertainty maps (Figs. 10b and 10e), UK-DM-LUR has regions with higher relative uncertainty than the UK-DM. This behavior is due to the addition of the microscale-LUR covariate, which increases the standard deviation associated with the regression model. In addition, some localized regions of high uncertainty can be observed in Fig. 10e. They are associated with passive dosimeters' locations and trafficked roads where the microscale-LUR covariate has caused an increase in $NO_2$ concentrations, rising the level of extrapolation in the regression model. The high uncertainty values in the upper left corner of 395 Figs. 10b and 10e correspond to the low $NO_2$ levels predicted in the Collserola mountains. These high uncertainty values can be reduced by considering the *Observatori Fabra* station, located in this area. However, as explained in Sect. 2.1, we excluded this station since its inclusion decreases the data-fusion model's ability to predict high $NO_2$ values in critical trafficked areas.

Regardless of the data-fusion method, the most polluted regions correspond to probabilities exceeding the annual limit above 0.7, as shown in Figs. 10c and 10f. When considering the UK-DM-LUR method, 13 % of the Barcelona municipality area has 400 0.7 or higher probabilities of exceeding the annual limit; and this percentage rises to 30 % when considering probabilities equal to or higher than 0.5. The *Eixample* district, which is the most polluted while being the most populous and densely

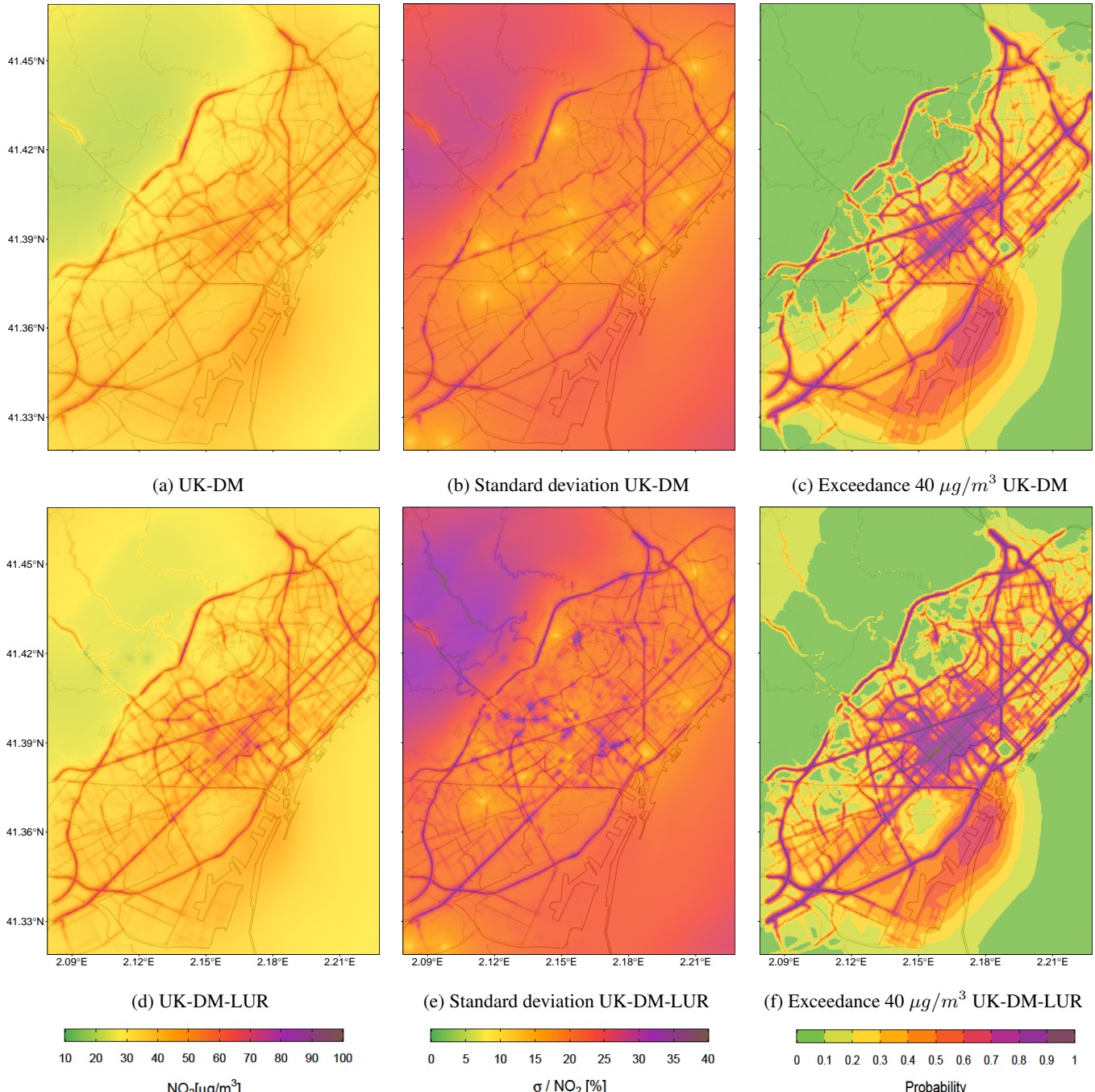

**Figure 10.** (a) NO$_2$ 2019 annual map resulting from applying UK-DM with the annual values, (b) Relative uncertainty associated with the predictions in (a), (c) Annual probability map of exceeding the 40 $\mu g/m^3$ NO$_2$ limit value using the values in (a) and (b), (d) NO$_2$ 2019 annual map resulting from applying UK-DM-LUR with the annual values, (e) Relative uncertainty associated to the predictions in (d), and (f) Annual probability map of exceeding the 40 $\mu g/m^3$ NO$_2$ limit value using the values in (d) and (e).

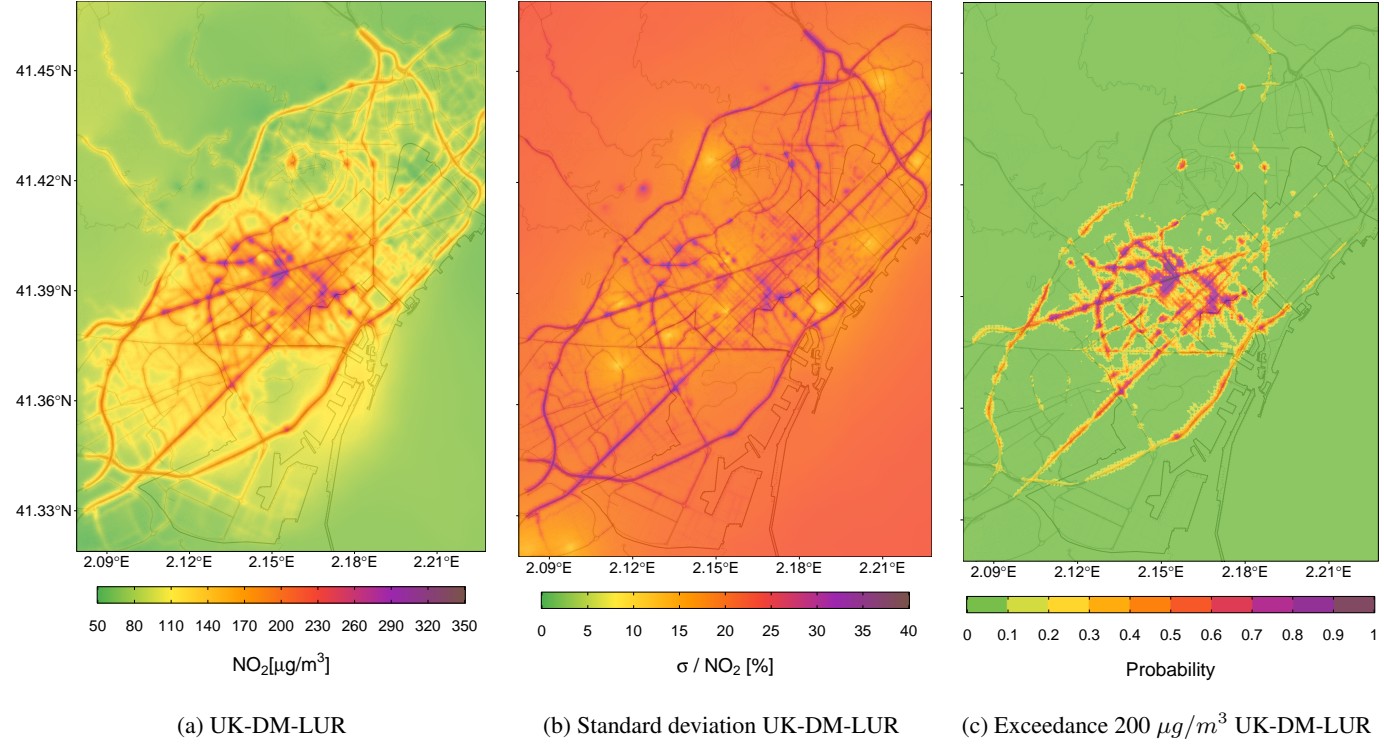

| (a) UK-DM-LUR | (b) Standard deviation UK-DM-LUR | (c) Exceedance 200 $\mu g/m^3$ UK-DM-LUR |

**Figure 11.** (a) Hourly NO$_2$ concentration map resulting of applying the UK-DM-LUR methodology at 9h UTC on 28/02/2019. (b) Relative uncertainty associated to the predictions in (a), (c) Hourly probability map of exceeding the 200 $\mu g/m^3$ NO$_2$ hourly averaged limit value using the UK-DM-LUR method at 9h UTC on 28/02/2019.

populated (approximately 270000 inhabitants and 36000 inhabitants per square kilometer (Ajuntament de Barcelona, 2019)), has 95 % of its area exceeding the annual limit with a probability equal to or higher to 0.5, and 69 % in the case of 0.7. Thus, significant evidence indicates that the annual legal limit was broadly exceeded in Barcelona in 2019. Stronger evidence could
405   be obtained by reducing the uncertainty associated with the results, either by a better correlated urban model or by increasing the monitoring system's coverage. To test a more restrictive threshold, we have analyzed the exceedance probability annual maps using the recommended WHO 2021 annual limit of 10 $\mu g/m^3$ (not shown here), obtaining probabilities above 0.9 over all the domain for both methodologies.

Figure 11 presents the NO$_2$ prediction at a specific hour, its associated relative uncertainty, and the exceedance probability
410   map based on the 200 $\mu g/m^3$ NO$_2$ hourly threshold (AAQD 2008/EC/50) for the UK-DM-LUR methodology. The goal is to illustrate that, apart from studying the long-term NO$_2$ mean values, the present methodology can also be used to correct short NO$_2$ exposure episodes such as the ones observed during traffic rush hours. Figure 11 corresponds to the peak traffic hour at 9 UTC on February 28, 2019, which was a particularly polluted hour reporting 138 and 201 $\mu g/m^3$ at the traffic monitoring stations of *Eixample* and *Gràcia*, respectively. Similar to Fig. 10, low uncertainty regions are obtained around the monitoring

stations' locations. Likewise, high relative uncertainty regions are associated with pollution hot-spots due to the extrapolation effect in the regression step. Concerning the exceedance probability maps shown in Fig. 11c , the city center and its major trafficked streets have the highest values (> 0.7). For the *Eixample* district, 19 % of the area exceeds the hourly limit with a probability equal to or higher than 0.5, and 6 % in the case of 0.7.

## 4    Conclusions

The present work assesses the added value of including a microscale-LUR basemap into a data-fusion method to obtain spatially bias-corrected urban maps of $NO_2$ at the hourly scale. To do so, we have compared two different data-fusion methods: (i) merging an urban dispersion model with the observational data of 12 monitoring stations using Universal Kriging (UK-DM), and (ii) adding to UK-DM a non-linear LUR model as a covariate in the Kriging workflow based on the GBM algorithm (UK-DM-LUR). The comparison is based on the statistical performance in LOOCV at each monitoring station, the resulting $NO_2$ maps, and their associated uncertainty.

The statistical performance of the microscale-LUR model has been assessed using a comprehensive nested CV. As expected, the obtained microscale-LUR basemap ($r = 0.64, RMSE = 11.87\,\mu g/m^3$) outperformed the raw annual-averaged dispersion model results ($r = 0.54, RMSE = 13.68\,\mu g/m^3$), highlighting the convenience of using passive dosimeters campaigns to explain the spatial distribution of $NO_2$. Moreover, a novel traffic density variable based on the combination of different traffic buffer sizes has been shown to have a significant influence (17.7 %) in the microscale-LUR basemap, suggesting its relevance in future microscale-LUR models.

Adding the microscale-LUR time-invariant spatial information (UK-DM-LUR) has been demonstrated to significantly improve the skills of the more straightforward data-fusion UK-DM method at the hourly scale, increasing the correlation coefficient ($r$) by 13 % and reducing the $RMSE$ by -24 % in average over all monitoring stations during 2019. Thus, our results suggest that data-fusion methods applied at the street-scale benefit from high-spatial resolution data such as passive dosimeters campaigns, urban morphology, or traffic intensity estimates. When only using monitoring stations in the data-fusion approach, the spatial patterns of $NO_2$ rely mainly on the urban model patterns. Generally, the better the temporal and spatial coverage of observational data, the better statistical performance can be achieved.

To check the consistency of the estimated uncertainty, we have empirically validated the UK-based uncertainties through a LOOCV. Despite the Universal Kriging's predicted variance is slightly overconfident and tends to degrade for extreme concentration values, we found that it is a meaningful estimate of the uncertainty. The PDFs of the error are close to the normal distribution, especially for the UK-DM-LUR approach. The spatial characterization of the uncertainty adds value to the $NO_2$ concentration maps, making data-fusion results more comprehensive for regulatory purposes, decision-makers, and health impact assessment. For instance, uncertainty maps can be used to allocate new observational stations or to plan future LCS campaigns. In this regard, our results show that pollution hot-spots are areas of high uncertainty underrepresented by the current monitoring system. We thus stress the need to monitor the vicinity of heavily trafficked roads better to increase the performance of data-fusion methods in predicting hourly and annual exceedances.

In developing our microscale-LUR model, a limitation arises when using campaigns conducted between February and March. Although the annualization adjustment factor corrects the $NO_2$ values, the spatial patterns are still linked to the period of the campaigns. If additional campaigns from different seasons of the year were available, assessing the seasonal bias effects on the spatial gradients would be highly interesting. Ideally, the basemap should be on a seasonal scale rather than a yearly scale. This highlights a potential improvement in our methodology that we could not quantify in the present analysis due to the lack of experimental campaigns during other seasons of the year. As another limitation, the *Observatori Fabra* station has been excluded from the data-fusion methodology because its inclusion worsened the results in the urban environment. Although its exclusion means losing relevant information regarding low $NO_2$-level areas, the primary objective of the urban model is to identify $NO_2$ exceedances in high-trafficked areas.

Local authorities frequently conduct air quality diagnoses solely based on available monitoring stations, resulting in inaccurate assessments of the situation since numerous local pollution hot-spots remain unmonitored. We have shown that data-fusion methods can provide a more comprehensive analysis by minimizing the sampling bias. For instance, in 2019, only the Gràcia and Eixample stations exceeded the annual legal $NO_2$ limit of 40 $\mu g/m^3$, and only four hourly exceedances were recorded during this period in Barcelona. In contrast, our results point out that large built-up areas and the main transit streets in the city recurrently exceeded the legal limits during the same period. Particularly, 13 % of Barcelona city has a probability of 0.7 or higher of exceeding the $NO_2$ annual limit value of 40 $\mu g/m^3$, which increases to 30 % with a probability of 0.5 or higher. For the Eixample district, which is the most populous and densely populated, those percentages are 69 % and 95 %, respectively.

A strong point of the presented methodology is the characterization of the $NO_2$ spatial patterns by combining two sources of information: the urban dispersion model and the microscale-LUR model. Therefore, the transferability of this method to other cities depends upon the existence of relevant passive dosimeter observations (or other observations providing constraints on the spatial variability at urban/street level) and the availability of a high-resolution urban air quality model. Regarding the urban dispersion model, key aspects are the availability of a detailed road network to derive meaningful emissions and utilizing a skilled regional model to prescribe the boundary conditions accurately. On the other hand, Appendix A presents an assessment of the necessary amount of samplers to retrieve a valid microscale-LUR model. On top of that, a network of monitoring stations plays a crucial role in the regression step of Universal Kriging, as a linear model is derived every hour. In this study, we observed that at least 4 monitoring stations have to be available to build robust linear regressions. However, this might vary depending on the specificities of the analysis, such as the urban model skills and the size of the city.

*Code and data availability.* The source code and the results, including the final kriging post-processed product (predicted concentrations, uncertainties, and exceedances), are publicly available via Zenodo on Criado et al. (2022). The xAire dosimeters campaign is publicly available in Perelló et al. (2021b). The input traffic data, coming from the bottom-up emission model HERMESv3 (Guevara et al., 2019), and the IDAEA-CSIC dosimeters campaign data (Benavides et al., 2019) are available upon request from the research group that developed them.

# Appendix A: Impact of selected passive dosimeter campaigns on the data-fusion results

An assessment of the passive dosimeters data needed for the present data-fusion methods is provided here. Although the specificities of the data, this assessment is intended to aid in the transferability to other cities. Firstly, Sect. A1 provides a statistical assessment of the data-fusion techniques as a function of the experimental campaign used. Secondly, Sect. A2 includes a brief discussion of the number of samplers required.

## A1   Impact of combining different experimental campaigns

We have calculated the effect of using campaigns from different years at two distinct levels: effects on the microscale-LUR performance, and effects on the overall data-fusion workflow performance (UK-DM-LUR).

### A1.1   Impact on the microscale-LUR performance

Applying the performance evaluation procedure described in Sect. 3.1.1, Table A1 compares statistical results for the microscale-LUR model when relying solely on data from the CSIC or the xAire campaigns. As a reference, we have also added the results of the raw CALIOPE-Urban model and the microscale-LUR performance when using both campaigns (already shown in Table 2).

| Campaign | Model | | n | COE | MB ($\mu g/m^3$) | r | RMSE ($\mu g/m^3$) |
|---|---|---|---|---|---|---|---|
| CSIC | | Training-validation set | 1580 | 0.51 | 0.24 | 0.85 | 8.70 |
| | Microscale-LUR | Test set without adding the residuals | 170 | 0.32 | 0.31 | 0.75 | 10.74 |
| | | Test set adding the residuals | 170 | 0.35 | -0.27 | 0.75 | 10.68 |
| | Raw CALIOPE-Urban | Annual mean | 170 | 0.20 | 0.71 | 0.67 | 12.66 |
| xAire | | Training-validation set | 6030 | 0.29 | -0.13 | 0.67 | 11.49 |
| | Microscale-LUR | Test set without adding the residuals | 660 | 0.23 | -0.18 | 0.59 | 12.40 |
| | | Test set adding the residuals | 660 | 0.26 | -0.25 | 0.64 | 11.87 |
| | Raw CALIOPE-Urban | Annual mean | 660 | 0.09 | -1.23 | 0.51 | 13.81 |
| CSIC and xAire | | Training-validation set | 7600 | 0.30 | 0.15 | 0.69 | 11.38 |
| | Microscale-LUR | Test set without adding the residuals | 840 | 0.24 | 0.22 | 0.62 | 12.17 |
| | | Test set adding the residuals | 840 | 0.27 | -0.27 | 0.64 | 11.87 |
| | Raw CALIOPE-Urban | Annual mean | 840 | 0.13 | -0.81 | 0.54 | 13.68 |

**Table A1.** Statistical results of the microscale-LUR model in nested CV, considering both campaigns or solely one of them. The 2017 annual mean concentration of $NO_2$ of the raw dispersion model (CALIOPE-Urban) is also shown.

The microscale-LUR model based solely on the CSIC campaign exhibits superior performance compared to the model based on both campaigns, whereas the model based solely on the xAire campaign demonstrates the opposite trend. However, there are notable differences in the number of data points and the motivation behind each campaign. The CSIC campaign deployed many fewer samplers (175), which raises concerns about possible overfitting. In this line, the *COE* statistic shows a significant

decline (~40 %) between the training set and the test set without residuals, although the decrease in performance for the other statistics is not as prominent. Additionally, we expect a higher data quality of the CSIC campaign, since it was conducted by a specialized research agency. In contrast, the xAire campaign was a citizen science initiative, involving school children and their families. All of this could have affected issues such as clustering (see Fig. 3), although the number of dosimeters of this campaign included here is considerably larger (669). Combining both campaigns allows us to consider more samples to characterize the complex $NO_2$ gradients in the city while reducing potential errors associated with overfitting and clustering.

### A1.2 Impact on the full data-fusion workflow performance

Figure A1 shows the statistical results (*COE*, *MB*, *r*, and *RMSE*) obtained through an hourly LOOCV approach across the 12 monitoring stations. The statistical analysis compares the Universal Kriging technique that employs only the CALIOPE-Urban output as a covariate (UK-DM), the Universal Kriging technique adding the microscale-LUR model resulting from combining both dosimeter campaigns (UK-DM-LUR), and the UK-DM-LUR models based only on one campaign (UK-DM-LUR CSIC and UK-DM-LUR xAire). For reference, the raw CALIOPE-Urban statistical results are also presented.

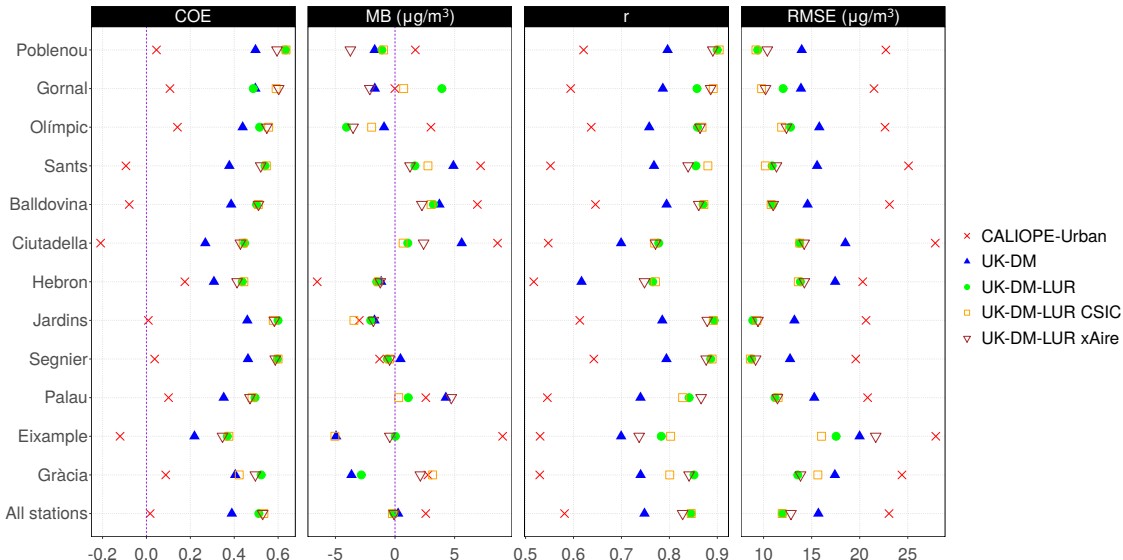

**Figure A1.** Statistical results for each station after applying UK-DM and UK-DM-LUR to 2019 hourly data in LOOCV. For the UK-DM-LUR application, we have considered developing the microscale-LUR model only with one experimental campaign (UK-DM-LUR CSIC or UK-DM-LUR xAire), or both of them (UK-DM-LUR). In addition, we show the statistical results for the CALIOPE-Urban estimates at each station. The *All stations* row refers to the average over all stations.

Regardless of the configuration, UK-DM-LUR improves the UK-DM methodology (and, therefore, CALIOPE-Urban) for the *COE*, *r*, and *RMSE* indicators. For the *MB* indicator, there is no clear trend once again. Once the microscale-LUR model

is integrated into the Universal Kriging framework, the statistical differences among UK-DM-LUR configurations were less significant than the ones shown in Table A1. It should be noted that the LOOCV is carried out in a limited number of monitoring stations (12), which represents a significant constraint on the current statistical evaluation. Despite this limitation in the evaluation, we consider that the broader spatial coverage of the samplers when combining both campaigns is the better option, allowing to capture a greater number of complex $NO_2$ structures not reproduced by CALIOPE-Urban.

## A2    Impact of the number of samplers considered on the microscale-LUR performance

For the case of using the two campaigns, we have computed the microscale-LUR performance gradually increasing the number of samplers from 140 to 790 by uniform increments of 50 random samplers, which results in 14 new models. In addition, we have also added the final model with all samplers (844) to make the comparison. To ensure the robustness of the results, we repeated these computations three times, randomly varying the selected samplers. Then, from these three series, the average and the standard deviation of the statistical indicators are computed. Figure A2 compares the *COE*, *MB*, *r*, and *RMSE* when gradually increasing the number of samplers for the training dataset, the test dataset, the test dataset interpolating the residuals, and the raw CALIOPE-Urban output.

As expected, as more samplers are considered, the standard deviation of the different metrics decreases. Also, an increasing trend in *COE* and *r* for the test sets is observed, while the same statistics decrease for the training sets. This opposite trend indicates that the overfitting is being reduced as more samplers are considered. For the test sets, the *RMSE* fluctuates around 12 $\mu g/m^3$ beyond 290 samplers with a moderated variability. Despite some fluctuations in the results, we can conclude that from 290 sampler onwards, the *COE* differences between training and test sets remain more or less constant, as well as the resulting *RMSE*. Therefore, based on these results, we would recommend a minimum of 290 samplers to build the microscale-LUR.

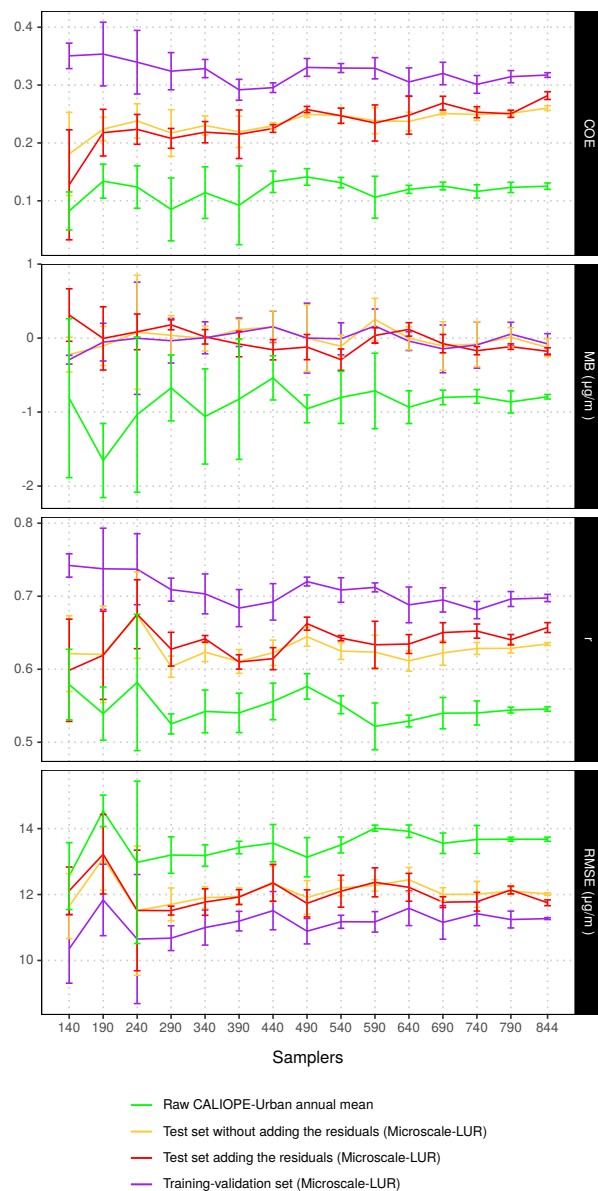

**Figure A2.** Statistical results of the 15 microscale-LUR models in nested CV. The models are built by considering both dosimeters campaigns and gradually increasing the number of samplers from 140 to 790 by uniform increments of 50 random samplers, in addition to the final model with all the samplers (844). The statistics represent the evaluation of the microscale-LUR models for the training and test (with and without the correction of the residuals) sets. The 2017 annual mean concentration of $NO_2$ of the raw dispersion model (CALIOPE-Urban) is also shown and evaluated in the dosimeter's locations.

*Author contributions.* AC implemented the data-fusion code and generated the figures. AC and JMA conducted the study and drafted the manuscript. JMA and JB processed the CALIOPE-Urban data. HP supported the validation of the microscale-LUR model. All authors contributed to the analysis and objectives of the document, and internally reviewed and supported the text.

*Competing interests.* The authors declare that they have no conflict of interest.

*Acknowledgements.* We acknowledge support from the *Ministerio de Ciencia, Innovación y Universidades* (MICINN) as part of the BROWN-ING project RTI2018-099894-BI00, from the VITALISE project (PID2019-108086RA-I00) funded by the MCIN/AEI/10.13039/501100011033, the MITIGATE project (PID2020-116324RA695 I00 / AEI /10.13039/501100011033) from the *Agencia Estatal de Investigación* (AEI), and the AXA Research Fund. The authors want to thank *Direcció General de Qualitat Ambiental i Canvi Climàtic - Generalitat de Catalunya* for providing observational data through the XVPCA, and IDAEA-CSIC for providing the experimental dosimeters campaign data. This project has also received funding from the European Union's Horizon 2020 research and innovation program under the Marie Skłodowska-Curie grant agreement H2020-MSCA-COFUND-2016-754433. BSC researchers thankfully acknowledge the computer resources at Marenostrum and the technical support provided by Barcelona Supercomputing Center (RES-AECT-2021-1-0027, RES-AECT-2021-2-0001).

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
