# Peer review of "Data fusion uncertainty-enabled methods to map street-scale hourly NO2 in Barcelona city: a case study with CALIOPE-Urban v1.0"

_EGUsphere, 2022_

## Author Response (AR2)

We are thankful for the numerous constructive comments and suggestions provided by the two reviewers. Below, the comments of the reviewers are indicated in *grey*, our answers in black, the modifications of the text after the comments of Anonymous Referee #1 in blue, and the modifications of the text after the comments of Anonymous Referee #2 in red.

**Anonymous Referee #1**

*Overview:*

*Criado et al. present a data-fusion workflow that uses Universal Kriging (UK) to merge dispersion model output (from CALIOPE-Urban) to hourly observations and microscale land-use regression (LUR) models. The authors' workflow is able to create high-resolution street-scale data of NO2 to compute exceedance probabilities, with uncertainty calculations based on the UK technique estimated error variance. This work is comprehensive and appears to have good improvement in correlation and error metrics. I will be happy to recommend this manuscript for publication after my (mostly minor) comments below are addressed.*

*Major comments:*

*1. I want to note that the code isn't available and thus cannot be reviewed in its current form. The authors state that "So, at this moment only reviewers can access these relevant sources under a previous mail in the request form." But the request form on Zenodo requires the full name, e-mail address, and affiliation of the requester, compromising the anonymity of the referees. Thus, I was not able to review the code that is associated with this work. I would request that the authors provide the code used in this work for review, either through the Editor or the GMD portal. While I appreciate that the authors have archived the code in a repository with a DOI, access during review is important. Many other authors have publicly archived their code when submitting to GMD, despite the manuscript being under review.*

We apologize for the problems associated with the revision of the code. The code is now publicly available in the following repository:

> Criado, A., Mateu Armengol, J., Petetin, H., Rodríguez-Rey, D., Benavides, J., Guevara, M., Pérez García-Pando, C., Soret, A., and Jorba, O. (2022). Code and data set from data fusion uncertainty-enabled methods to map street-scale hourly NO2 in Barcelona city: a case study with CALIOPE-Urban v1.0 (1.0). Zenodo, https://doi.org/10.5281/zenodo.7185913

*2. The authors have performed data-fusion using data from two LCS campaigns, one from 2018, and one from 2017. Is there an impact on the quality of the data-fusion technique if LCS data are provided in different years? Similarly, if only one LCS campaign data set is used, how would it impact the quality of the results? A brief assessment of how much data is necessary and the applicability of the methods shown in this work will help future readers to apply this technique in the future to other major cities where urban pollution is also a major health issue.*

We agree with the reviewer that assessing the data-fusion skills depending on the available data would help future researchers apply this technique. An Appendix has been added to the new version of the manuscript :

[revised manuscript text omitted]

Regarding transferability to other cities, and in line with *Specific comment 6* of Anonymous Referee #2, the following text has been added to Section *4. Conclusions*:

> L464-473: A strong point of the presented methodology is the characterization of the NO2 spatial patterns by combining two sources of information: the urban dispersion model and the microscale-LUR model. Therefore, the transferability of this method to other cities depends upon the existence of relevant passive dosimeter observations (or other observations providing constraints on the spatial variability at urban/street level) and the availability of a high-resolution urban air quality model. Regarding the urban dispersion model, key aspects are the availability of a detailed road network to derive meaningful emissions and using a skilled regional model to prescribe the boundary conditions accurately. On the other hand, Appendix A presents an assessment of the necessary amount of samplers to retrieve a valid microscale-LUR model. On top of that, a network of monitoring stations plays a crucial role in the regression step of Universal Kriging, as a linear model is derived every hour. In this study, we observed that at least 4 monitoring stations have to be available to build robust linear regressions. However, this might vary depending on the specificities of the analysis, such as the urban model skills and the size of the city.

*Specific comments:*

*1. L57: "while the time-dependent LCS network explains the temporal behavior." Is it implied that the temporal behavior is short-term here in contrast with the long-term spatial distribution provided by the urban model?*

Thank you for raising this point. Indeed, *temporal behavior* does refer to short-term temporal behavior. We have clarified this point in the new version of the manuscript as follows:

> L54-57: Schneider et al. (2017) use a popular geostatistic technique, Universal Kriging, considering the time-aggregated annual mean of an urban model as a basemap (or climatology) to explain the long-term spatial gradients at the street-scale, while the time-dependent LCS network explains the short-term temporal behavior.

> L132-134: While the monitoring stations and the urban dispersion model provide information on the pollutants' short-term temporal behavior, the microscale-LUR basemap (long-term mean) remains constant in time.

*2. L134: An adjustment factor is computed as the ratio between the observed 2017 annual mean and the average over the period of the experimental campaign. The LCS campaigns span only a few weeks (February 16th to March 15th, 2018; and February and March, 2017) – why is the 2017 annual mean used here instead of, for example, February- March mean?*

The main reason to scale the campaigns at the annual level is that the resulting basemap is used to correct the model output throughout the whole year. However, the basemap may better represent the NO2 gradients observed during February-March than the real annual mean gradients. Ideally, if several campaigns at different seasons were available, a microscale-LUR model would have been fitted for each season. In the new version of the manuscript, we have clearly stated this limitation in the following lines:

> L145-148: Note that the microscale-LUR model is trained using experimental campaigns deployed in February and March. As a result, even though the

annualization process corrects the NO2 levels and the predictors are expressed as annual averages, the captured spatial gradients may still have a significant seasonal bias.

L447-452: In developing our microscale-LUR model, a limitation arises when using campaigns conducted between February and March. Although the annualization adjustment factor corrects the NO2 values, the spatial patterns are still linked to the period of the campaigns. If additional campaigns from different seasons of the year were available, assessing the seasonal bias effects on the spatial gradients would be highly interesting. Ideally, the basemap should be on a seasonal scale rather than a yearly scale. This highlights a potential improvement in our methodology that we could not quantify in the present analysis due to the lack of experimental campaigns during other seasons of the year.

3. *L136-137: The authors say that this processing adds some noise to experimental results but corrects the "environmental conditions influence". What environmental conditions are referred to here? My impression is that this would mainly correct for bias in the low cost sensors' instruments.*

Environmental conditions (e.g., wind speed, atmospheric stability, precipitation, radiation, temperature) likely induce a difference between the NO2 levels averaged over the campaign period and the annual mean. We agree with the reviewer that the adjustment factor mainly corrects the bias of the campaign-averaged levels with respect to the annual mean. We have clarified this point in the manuscript as follows:

L142-145: Despite adding some noise to the experimental results, it corrects the bias induced by environmental conditions (e.g., wind speed, atmospheric stability, precipitation, radiation, temperature) and also  allows combining both campaigns, producing a dataset of 844 samplers on which the microscale-LUR model relies.

4. *Figure 3: Useful to label inset in the figure "Combined data" "CSIC" "xAire".*

As suggested by Anonymous Referee #1, we have included titles in each of the three panels to easily identify the different campaigns. Figure 3 in the manuscript has been changed by the following one:

[Figure]

**Figure 3.** *Sampler locations of the two different NO2 experimental campaigns used to train the microscale-LUR model. The left panel shows the NO2 values and the locations of the combined campaigns. The top- and bottom-right panels show the CSIC and xAire campaign locations, respectively. The colour scale refers to the 2017 annualized NO2 values, in µg/m3. The map has been generated using ggplot2 and ggmap R packages, and data from OpenStreetMap. © OpenStreetMap contributors 2017. Distributed under the Open Data Commons Open Database License (ODbL) v1.0.*

> 5. *L247-248: The authors indicate that with the criteria (covariate slopes must be positive, less than four observations available in the hour) 14% and 2% of the hours in UK-DM, UK- DM-LUR are not corrected. How much percentage of these are due to negative covariate slopes? And how much are due to too few observations? If there is a significant percentage of nonphysical negative covariates, is there a common pattern to the conditions causing these?*

We agree with Anonymous Referee #1 that detailing the results of the statistical check is relevant information to add to the paper. In the new version of the manuscript, the following text has been added in Lines 258-264:

> L258-264: Following the above criteria, the percentage of cases with fewer than 4 monitoring observations is relatively small, 0.034 % (3 hours), and is the same for each kriging application. For the UK-DM methodology, 14.11 % of the hours have not been corrected due to negative regression coefficients. On the other hand, for the case of UK-DM-LUR, only  1.47 % of the hours have been discarded due to a negative regression coefficient in both covariates.  As Benavides et al. (2019) identified, the poor skills of the urban model are attributed to low wind speeds and atmospheric stability situations, for which the performance of the mesoscale model decreases. Concerning the static microscale-LUR basemap, the

poor correlation on an hourly basis is associated with hours that significantly deviate from the average behavior.

*6. L273: "We attribute this behavior … also to the already poor predictive skills of CALIOPE- Urban in this concentration range." A citation will be useful for CALIOPE-Urban's underperformance in high-NOx conditions.*

We forgot to refer to Fig. 5a in which the referred behavior of CALIOPE-Urban can be observed. The new version of the manuscript clarifies this point as follows:

L291-293: We attribute this behavior to the limited number of points in this range, which can weaken the model training, particularly in the nested CV context, but also to the already poor predictive skills of CALIOPE-Urban in this concentration range as seen in Fig. 5a.

*Technical corrections:*
*1. L204: "back-transformed" -> "back-transformation"*
*2. L210: "exceedance (P) a certain…" -> "exceedance (P) of a certain…"*
*3. Lines 233, 234: middle dot -> cross sign for scientific notation.*
*4. Figure 7: Units for MB, RMSE are missing the "^3" (shows as micrograms/m)*

We are thankful for the technical corrections provided by Anonymous Referee #1. All of them have been included in the text as suggested.

**Anonymous Referee #2**

*Overview:*

*Urban NO2 pollution shows strong gradients, and is almost always undersampled by reference networks. The authors present an interesting approach to assimilate complementary observational datasets, having different temporal sampling and accuracy, in a high-resolution urban dispersion model. Using Universal Kriging, they show that the predictive performance of the dispersion model improves if hourly measurements are included. Moreover, they show that the system further improves if a basemap is added in the data fusion, based on 840 Palmes tube measurements. The observations clearly resolve local spatial structures which are not properly described by the street model alone. The authors show that the error margin provided from the Kriging method is fairly realistic, which enables them to calculate maps with expected local exceedances of annual and hourly limit values of NO2 air pollution.*

*The paper is well-referenced, positioning the study well in the current research efforts on this topic. I recommend publication after addressing the following comments:*

*General comments:*

*1.     The microscale LUR model is trained by Palmes observations done in end-February/begin-March. Although an annualization is applied (L132-L136), I assume the resulting basemap would look differently when evaluated for months with e.g. different typical NOx lifetime, boundary layer height, dominant wind direction. Does the usage of a February/March basemap throughout the year introduce a significant seasonal bias?*

Anonymous Referee #2 pointed out that the microscale-LUR model was trained using experimental campaigns conducted between February and March. Although an adjustment factor corrects the seasonal bias of the dosimeter values through the annualization process, the spatial patterns reflected by the dosimeters are still linked to that particular season. Consequently, there may be a stationary bias. However, with the currently available campaigns, we believe that annualizing the values is the best way to use this information beyond the sampling period to obtain a basemap for the entire year.

This discussion highlights a limitation of our methodology due to the lack of data throughout the year. We have included it to enrich the manuscript in Sections 2.3 and 4. Note that this question is also linked with *Specific comment 2* of Anonymous Referee #1:

> L145-148: Note that the microscale-LUR model is trained using experimental campaigns deployed in February and March. As a result, even though the annualization process corrects the NO2 levels and the predictors are expressed as annual averages, the captured spatial gradients may still have a significant seasonal bias.

> L447-452: In developing our microscale-LUR model, a limitation arises when using campaigns conducted between February and March. Although the annualization adjustment factor corrects the NO2 values, the spatial patterns are still linked to the period of the campaigns. If additional campaigns from different seasons of the year were available, assessing the seasonal bias effects

on the spatial gradients would be highly interesting. Ideally, the basemap should be on a seasonal scale rather than a yearly scale. This highlights a potential improvement in our methodology that we could not quantify in the present analysis due to the lack of experimental campaigns during other seasons of the year.

*2.  The study uses hourly NO2 measurements of 12 reference stations in the Barcelona area. To my knowledge, NO2 reference measurements are also done at the Observatori Fabra site, which is also within the considered domain. Is there any reason why this station is excluded?*

We thank the reviewer for raising this point. Indeed, data from the Observatori Fabra station is available and located within the study domain. However, after conducting statistical analyses, we decided to exclude this station. In the new version, we added the station's location in Fig. 2 and the text below describing why it was excluded:

[Figure]

**Figure 2.** *Domain of study and location of the referenced monitoring stations. The map has been generated using ggplot2 and ggmap R packages, and data from OpenStreetMap. © OpenStreetMap contributors 2017. Distributed under the Open Data Commons Open Database License (ODbL) v1.0. Map tiles are © Stamen Design, under a Creative Commons Attribution (CC BY 3.0) license.*

L93-94: There are  13 stations available on the Barcelona agglomeration (Fig. 2), with a percentage of availability of hourly data greater than 93 %.

L94-96: Gràcia and Eixample are urban traffic monitoring stations, Segnier, Observatori Fabra and Jardins are sub-urban background stations, and the remaining 8 correspond to urban background stations.

L96-101: The Observatori Fabra station is not used in our data-fusion methodology since its inclusion significantly degraded the data-fusion skills in the urban environment. This is expected since the station is located on a hill

relatively far from built-up areas. In fact, it is not exactly an urban station because it measures air pollution above the urban canopy while the other stations measure pollution within the urban canopy. We are aware that by removing this station, we may lose relevant information on the low NO2-level regions surrounding the city. However, the main goal of our urban model is to characterize NO2 exceedances in critical trafficked areas. Therefore, we decided to exclude the Observatori Fabra station.

L393-396: The high uncertainty values in the upper left corner of Figs. 10b and 10e correspond to the low NO2 levels predicted in the Collserola mountains. These high uncertainty values can be reduced by considering the Observatory Fabra station, located in this area. However, as explained in Sect. 2.1, we excluded this station since its inclusion decreases the data-fusion model's ability to predict high NO2 values in critical trafficked areas.

L452-455: As another limitation, the Observatori Fabra station has been excluded from the data-fusion methodology because its inclusion worsened the results in the urban environment. Although its exclusion means losing relevant information regarding low NO2-level areas, the primary objective of the urban model is to identify NO2 exceedances in high-trafficked areas.

*3.    Oftentimes, local authorities evaluate the air quality in their city based on measurements of the reference network only. This gives a distorted impression, as many local exceedances are not sampled. Data fusion methods, such as in this study, correct for this sampling bias. The authors show that a large part of the city does not meet the annual and hourly limit values for NO2 (L365-385). It would be interesting to see how this contrasts with an analysis based on station data alone.*

We thank the reviewer again for highlighting an advantage of using our methodology that we did not explicitly refer to in our original manuscript. Indeed, Munir et al. (2021) also raised this point. They compared various data-fusion methods incorporating observational NO2 data from multiple sources (monitoring stations, diffusion tubes, and low-cost sensors). In line with the reviewer's comment, they showed that monitoring stations provide the most accurate air quality information. However, an analysis based on station data alone fails to capture the spatial patterns and exceedances across the urban area needed for exposure studies. In this context, we believe that adding the following comment to the conclusions will increase the impact of our results:

L456-463: Local authorities frequently conduct air quality diagnoses solely based on available monitoring stations, resulting in inaccurate assessments of the situation since numerous local pollution hot-spots remain unmonitored. We have shown that data-fusion methods can provide a more comprehensive analysis by minimizing the sampling bias. For instance, in 2019, only the Gràcia and Eixample stations exceeded the annual legal NO2 limit of 40 µg/m3, and only four hourly exceedances were recorded during this period in Barcelona. In contrast, our results point out that large built-up areas and the main transit streets in the city recurrently exceeded the legal limits during the same period. Particularly, 13% of Barcelona city has a probability of 0.7 or higher of exceeding the NO2 annual limit value of 40 µg/m3, which increases to 30% with a probability of 0.5 or higher. For the Eixample district, which is the most populous and densely populated, those percentages are 69% and 95%, respectively.

*Specific comments:*

*4.       Section 3.1.2, Figure 6: The LUR basemap (6a) seems to be richer in detail than the mean of the dispersion model (6b). However, it misses the lower pollution levels in the mountainous area in the NW part of the domain. This introduces an unwanted bias in this area for UK-DM-LUR when compared to UK-DM, which escapes the validation statistics as there is no reference data available (or used) in this area. How could the microscale-LUR model be improved for non-built-up areas?*

We agree with the reviewer that the microscale-LUR basemap overpredicts the upper left corner of the domain, where NO2 levels are low. Moreover, as the referee commented, this issue is not captured in the statistical evaluation since we excluded the Observatori Fabra station. This unwanted bias is caused by the lack of samplers in this region and, more generally, the lack of samplers in low NO2-level areas. Thus, to improve our microscale-LUR model in the non-built-up areas, we would require more passive dosimeters in low NO2 regions. Alternatively, we could assign more weight to the samplers in these areas during the microscale-LUR model training process. However, this would probably degrade results in the built-up areas. This bias in the low NO2-level areas has been highlighted in the new version of the manuscript. We have clearly stated that this question escapes the statistical evaluation of the data-fusion method:

> L313-318: For instance, there is a noticeable increase in NO2 levels for the microscale-LUR basemap (Fig. 6a) in the mountainous north-western area of the study domain. This artifact is probably caused by the spatial distribution of the passive dosimeters campaigns (Fig. 3), which poorly cover this region. The NO2 overprediction of this area is not reflected in the statistical evaluation of the data-fusion since we deliberately omitted the monitoring station located in this area. We excluded this station to improve the data-fusion model's ability to capture NO2 exceedances in built-up areas, which is the main goal of the urban model.

*5.       Section 3.2.2, Figure 8: Can part of the skewness of the distributions be explained from the back-transformation from the log-domain (L202-204)?*

The log-transformation is applied to correct the skewness of the NO2 distribution before applying Kriging. As pointed out by the reviewer, when back-transforming the results, a skewed distribution is retrieved. The skewness of the distribution of observed NO2, which consists of recurrent moderate values and infrequent peaks, is linked to the distribution of the normalized bias shown in Fig. 8. We have clarified this point in the new version when discussing Fig. 8:

> L363-367: Both methodologies, especially UK-DM, exhibit negative skewness. This is because the corrected model struggles to capture the infrequent high pollution peaks, tending to underestimate them significantly. Thus, negative biases (Mh<Oh) are rare but stronger. On the other hand, the model tends to overpredict moderate observed values slightly. Therefore, positive biases (Mh>Oh) are more frequent and less severe. In agreement with the overall null bias, the rare strong underestimations are compensated by frequent moderate overestimations.

*6.       Conclusions: I miss some general words about transferability of this method to other cities, referring to dependencies on databases, dispersion models, and local*

*network configurations. Also, I recommend summarizing briefly the estimated exceedances in L265-385.*

We have added some information about transferability in the conclusions. This is also linked with the *Major comment 2* of Anonymous Referee #1:

> L464-473: A strong point of the presented methodology is the characterization of the NO2 spatial patterns by combining two sources of information: the urban dispersion model and the microscale-LUR model. Therefore, the transferability of this method to other cities depends upon the existence of relevant passive dosimeter observations (or other observations providing constraints on the spatial variability at urban/street level) and the availability of a high-resolution urban air quality model. Regarding the urban dispersion model, key aspects are the availability of a detailed road network to derive meaningful emissions and utilizing a skilled regional model to prescribe the boundary conditions accurately. On the other hand, Appendix A presents an assessment of the necessary amount of samplers to retrieve a valid microscale-LUR model. On top of that, a network of monitoring stations plays a crucial role in the regression step of Universal Kriging, as a linear model is derived every hour. In this study, we observed that at least 4 monitoring stations have to be available to build robust linear regressions. However, this might vary depending on the specificities of the analysis, such as the urban model skills and the size of the city.

We have also summarized the exceedances in the conclusions:

> L461-463: Particularly, 13% of Barcelona city has a probability of 0.7 or higher of exceeding the NO2 annual limit value of 40 µg/m3, which increases to 30% with a probability of 0.5 or higher. For the Eixample district, which is the most populous and densely populated, those percentages are 69% and 95%, respectively.

*7. L394-395: "The obtained microscale-LUR basemap (r=0.64, RMSE=11.87µg/m3) outperformed the raw annual-averaged dispersion model results (r=0.54, RMSE=13.68 µg/m3)". This is not very surprising, as the annual-averaged model results is one of the inputs of the LUR model.*

Thanks for your comment, we have modified that part to reflect the fact that the microscale-LUR basemap outperforms the raw annual-averaged dispersion model results is indeed not surprising:

> L425-427: As expected, the obtained microscale-LUR basemap (r=0.64, RMSE=11.87 µg/m3) outperformed the raw annual-averaged dispersion model results (r=0.54, RMSE=13.68 µg/m3), highlighting the convenience of using passive dosimeters campaigns to explain the spatial distribution of NO2.

*Technical corrections*

We are thankful for the technical corrections provided by the reviewer. All suggested modifications are changed in the new version of the manuscript.

*L22: "obtaining high-resolution exposure to NO2 is crucial"→ "obtaining information on high-resolution exposure to NO2 is crucial"*

L22-23: In this context, obtaining information on high-resolution exposure to NO2 is crucial for decision-making in urban air quality management.

*L245: "if their slope is positive". Confusing for me. I guess that a positive slope refers to a positive coefficient in the linear combination.*

L250: The correlation coefficient (r) and the regression coefficient (slope) of the regression model…
L255: the covariates are used only if their regression coefficient is positive…
L256: In case all regression coefficients are negative…